

# Propagating information from snow observations with CrocO ensemble data assimilation system: a 10-years case study over a snow depth observation network

Bertrand Cluzet[1], Matthieu Lafaysse[1], César Deschamps-Berger[1, 2], Matthieu Vernay[1], and Marie Dumont[1]

[1]Univ. Grenoble Alpes, Université de Toulouse, Météo-France, CNRS, CNRM, Centre d'Études de la Neige, Grenoble, France
[2]Centre d'Etudes Spatiales de la Biosphère, CESBIO, Univ. Toulouse, CNES/CNRS/INRA/IRD/UPS, 31401 Toulouse, France

**Correspondence:** Bertrand Cluzet (bertrand.cluzet@meteo.fr)

**Abstract.** The mountainous snow cover is highly variable at all temporal and spatial scales. Snowpack models only imperfectly represent this variability, because of uncertain meteorological inputs, physical parameterisations, and unresolved terrain features. In-situ observations of the height of snow (HS), despite their limited representativeness, could help constrain intermediate and large scale modelling errors by means of data assimilation. In this work, we assimilate HS observations from

an in-situ network of 295 stations covering the French Alps, Pyrenees and Andorra, over the period 2009-2019. In view of assimilating such observations into a spatialised snow cover modelling framework, we investigate whether such observations can be used to correct neighbouring snowpack simulations. We use CrocO, an ensemble data assimilation framework of snow cover modelling, based on a Particle Filter suited to the propagation of information from observed to unobserved areas. This ensemble system already benefits from meteorological observations, assimilated within SAFRAN analysis scheme. CrocO also

proposes various localisation strategies to assimilate snow observations. These approaches are evaluated in a Leave-One-Out setup against the operational deterministic model and its ensemble open-loop counterpart, both running without HS assimilation. Results show that intermediate localisation radius of 35-50 km yield a slightly lower root mean square error (RMSE), and a better Spread-Skill than the strategy assimilating all the observations from a whole mountain range. Significant continuous ranked probability score (CRPS) improvements of about 13% are obtained in the areas where the open-loop modelling errors

are the largest, e.g. the Haute-Ariège, Andorra and the Extreme Southern Alps. Over these areas, weather station observations are generally sparser, resulting in more uncertain meteorological analyses, and therefore snow simulations. In-situ HS observations thus shows an interesting complementarity with meteorological observations to better constrain snow cover simulations over large areas.



## 1   Introduction

Better monitoring the spatio-temporal variability of the mountainous snow cover is paramount to improve the forecasting of snow-related hazards (Morin et al., 2020) and anticipate downstream river flow (Lettenmaier et al., 2015). In mountainous terrain, the snow cover inherits a high spatial variability from several factors. The topography controls on the precipitation phase, air temperature, wind exposition and radiation fluxes (Durand et al., 1993; Oliphant et al., 2003). Wind drift redistributes snow at every scale (Mott et al., 2018). Finally, vegetation traps the snow (Sturm et al., 2001) and also affects its net shortwave

and longwave radiation (Qu and Hall, 2014; Malle et al., 2019).

Snowpack models are commonly used to derive snowpack properties in the mountains. Yet, their ability to represent snow cover variability over large areas is inherently limited by large errors in their meteorological forcings (Raleigh et al., 2015), and uncertain physical parameterisations (Essery et al., 2013; Krinner et al., 2018). In addition, explicitly accounting for processes such as wind drift and snow-vegetation interaction is not yet affordable at large scales.

In that context, additional sources of information are needed to mitigate snowpack modelling uncertainty in the mountains. Information from weather stations located in the mountains provide better estimates of surface meteorology than Numerical Weather Prediction (NWP) models. Dedicated downscaling and analysis schemes such as SAFRAN (Durand et al., 1993) or RhiresD interpolation in Switzerland (Frei and Schär, 1998) can be used to efficiently reduce the large errors of the NWP models in the mountains, in particular by the assimilation of local precipitation observations. Such approaches significantly

improve snow cover simulations (Durand et al., 1999; Magnusson et al., 2014). These weather stations, however, are generally located below 1200m (Frei and Schär, 1998; Vernay et al., in prep), and important errors in precipitations (for example) remain at higher elevations (Magnusson et al., 2014).

Data assimilation of snowpack observations, may help address this issue in complement to these observations. Remotely-sensed retrieval of snow bulk properties (e.g. the height of snow (HS, m) and the snow water equivalent (SWE, $\mathrm{kg\,m^{-2}}$)) is a promising

wealth of snowpack observations for data assimilation (e.g. Margulis et al., 2019) but it is inherently limited by spatio-temporal gaps (De Lannoy et al., 2012). In-situ observations of HS and SWE cover large mountainous areas and are operational on a daily basis in numerous countries (e.g. Serreze et al., 1999; Jonas et al., 2009; Durand et al., 2009b; Cantet et al., 2019). Their potential to improve local simulations is unambiguous as demonstrated by many studies (e.g. Magnusson et al., 2017; Piazzi et al., 2018; Smyth et al., 2019; Cantet et al., 2019). However, the representativeness of such observations is limited by the

snow cover spatial variability (Grünewald and Lehning, 2015; Lejeune et al., 2019). The potential to transfer information into neighbouring areas is therefore a key question when considering their potential added value for snow cover modelling over large domains (e.g. Slater and Clark, 2006; Gichamo and Tarboton, 2019). This question has long been debated. Cantet et al. (2019) successfully applied a spatialised Particle Filter (PF) over a very large domain (Southern Quebec), and with a loose observation network, though not in a rugged terrain, i.e. less spatial variability. In alpine terrain, Magnusson et al. (2014);

Winstral et al. (2019) showed that enhancing snow cover simulations with in-situ snow observations from a dense network in Switzerland reduced modelling errors over unobserved locations. It is yet to demonstrate that this approach can be applied over mountainous areas with a coarser in-situ observational coverage (Largeron et al., 2020).





Here, we investigate whether the assimilation of in-situ HS observations can improve simulations of Météo-France operational modelling chain for snow cover monitoring and avalanche hazard forecasting in the vicinity of the measurement stations, and
what is the most appropriate assimilation strategy for that purpose. We assess this in a network of in-situ HS observations over the French Alps, French Pyrenees, and Andorra, with contrasted observation densities. We use CrocO, an ensemble data assimilation system of snow cover modelling (Cluzet et al., 2021). CrocO is built around an ensemble version of the operational modelling system of Météo-France (Vionnet et al., 2012; Vernay et al., in prep), accounting for modelling uncertainties from the meteorological forcings (Charrois et al., 2016; Deschamps-Berger et al., in review) and the snowpack model itself (Lafaysse
et al., 2017; Dumont et al., 2020). CrocO includes several versions of the Particle Filter (PF) tailored for the propagation of information from observed into unobserved areas (Cluzet et al., 2021). These variants are used in a localised framework, in which only observations coming from a certain radius around the considered location are assimilated (Penny and Miyoshi, 2016; Poterjoy, 2016; Farchi and Bocquet, 2018). To assess the potential transfer of information, we opt for a Leave-One-Out approach (LOO) (e.g. Slater and Clark, 2006), whereby on each date, local observations are excluded from the assimilation
but kept for evaluation. If such potential transfer could be demonstrated, it would mean that the assimilation method is able to improve simulations at a sufficient distance of available observations to be efficient over the whole simulation domain. In other words, this network of observations could be used to constrain spatialised snowpack simulations over the French Alps, Pyrenees and Andorra. Furthermore, the methodology could be applied to other areas with similar densities of observations. To summarize, the following questions will be addressed in this paper:

– What is the performance of data assimilation compared with the operational and ensemble models?

   – Does data assimilation manage to propagate information in space?

   – What is the best localisation strategy for assimilation?

   – Could an increased observation density yield better results for assimilation?

The study area, observations, modelling chain and data assimilation scheme are described in Sec. 2. In Sec. 3, the evaluation
strategy and scores are presented. The results are presented and discussed in Sec. 4 & 5. We finally conclude and open research perspectives in Sec. 6.

## 2 Material and methods

### 2.1 Study area and observations

The study area spans the French sides of the Alps and Pyrenees and Andorra. The French Alps culminate at the Mont-Blanc
(4810 m) and are higher and about two times larger than the French Pyrenees (culminating at Vignemale, 3298 m). Andorra is a principality located at the center-East of the Pyrenees. In the following, for the sake of simplicity, we will refer to French Pyrenees and Andorra as "Pyrenees", and to French Alps as "Alps".





The winter climate of the Alps is contrasted between the North and the South. The Southern Alps are on average drier than the Northern (Isotta et al., 2014). The Pyrenees are very elongated with a strong longitudinal gradient between the humid oceanic Western side to the drier Mediterranean Eastern side. The elevation of the winter snow line is around 1500 m in the Pyrenees (Durand et al., 2012), and about 1200 m in the Northern Alps (Durand et al., 2009a). Finally, the inter-annual variability of the snow cover is marked in both massifs (Durand et al., 2009a; Gascoin et al., 2015).

In this work, we perform snowpack simulations in a network of 295 daily HS observations stations. 217 stations are located in the Alps, and 78 in the Pyrenees (of which 7 are in Andorra). This network is an aggregate of several data sources. Most of the observations (144 stations) come from ski resorts, where HS is manually observed every morning during the commercial season (mid-December to April in general). The second source is a network of climatological observations (77 stations) in which several meteorological parameters and HS are observed on a daily basis for the whole year. These stations are generally located around populated areas or in ski resorts. A few sites (19 stations) come from various automated measurements in ski resorts. Two networks of automated HS sensors were also used: Météo-France's Nivôses (27 stations) and Électricité de France (EDF) EDFNIVO stations (28 stations), the latter only from the winter season 2016-2017 on. These networks are located in remote areas and at generally higher altitudes than the rest of the observations.

The density of observations within each SAFRAN massif (Fig. 1) is very variable, from less than 0.5 daily observations per hundred $\mathrm{km}^2$ in the Extremely Southern Alps and Western Pyrenees to more than ten times higher densities in the Mont-Blanc massif . It is mainly explained by the variable density of ski resorts. Although the density of observations is generally lower than in the Alps, the Pyrenees exhibit two clusters of dense observations, in the Central Western part around Bigorre and in the Central eastern part close to Andorra. In the Alps, the density of observations is especially high from the Northern to the South Central area. The Southern massifs, as well as the lower altitude western massifs generally have less observations.

Fig. 2 shows the histograms of the available daily observations per 300 m-elevation bands for each year, together with the number of stations with at least one observation. A notable increase in the observations count above 2100 m for the three last years can be explained by the inclusion of the EDFNIVO stations. Fig. 3 shows the number of observation per month. It increases from 3000 during Fall to 6000 in January-March (when the ski resorts are open), suggesting that the beginning and end of season are less well observed both in terms of number of observations and spatial coverage.

## 2.2 Ensemble data assimilation setup

The ensemble system consists in an ensemble of meteorological forcings generated by stochastic perturbations, forcing a multiphysics ensemble of snow models as described in Cluzet et al. (2020) and Cluzet et al. (2021). The total number of ensemble members was set to 160. An open-loop run (i.e. without assimilation) was performed to serve as reference. Only a few changes



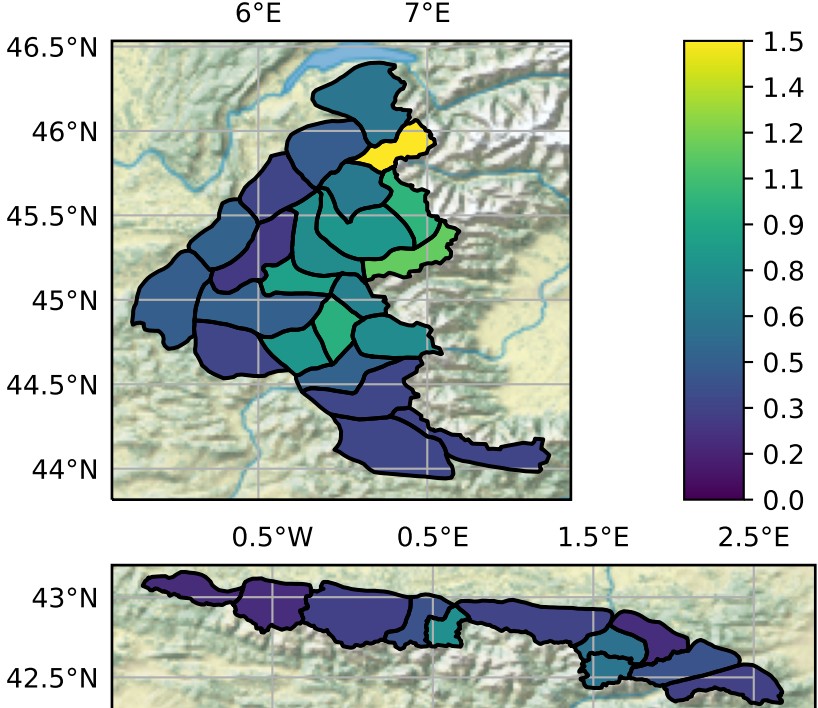

**Figure 1.** Average daily observation density (per 100 km²) within each SAFRAN massif, in the French Alps (top panel) and French Pyrenees/Andorra (bottom panel).

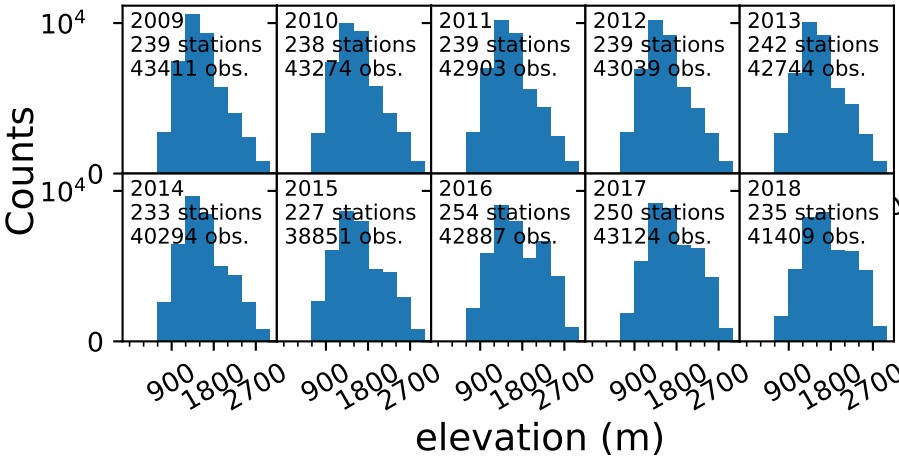

**Figure 2.** Number of daily observations per 300 m elevation bands for each winter season over the whole domain. The number of available stations is indicated in the caption.


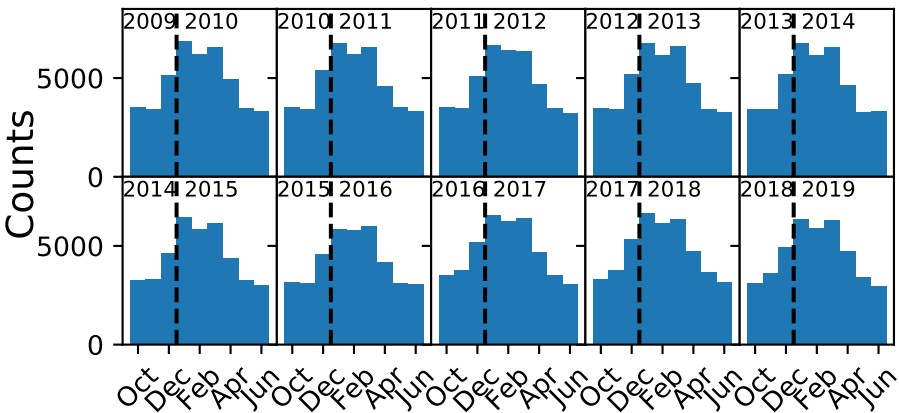

**Figure 3.** Monthly observation count for each winter year over the whole domain.

were performed in the ensemble setup, which are described in Secs. 2.2.1 and 2.2.2.

### 2.2.1  Ensemble of snowpack models

The simulation setup is based on a multiphysics framework representing the uncertainties of the main physical parameteri-
sations of Crocus (Lafaysse et al., 2017; Cluzet et al., 2020). However, in this paper, the advanced radiative transfer scheme
TARTES (Libois et al., 2013, 2015) was not used contrary to previous studies (Cluzet et al., 2020, 2021) because it requires
Light Absorbing Particles (LAP) fluxes from chemistry transport models such as MOCAGE, ALADIN or GFDL_AR4 (Josse
et al., 2004; Nabat et al., 2015; Horowitz et al., 2020). To date, such products are not interpolated within SAFRAN geometry
and would require a specific treatment and validation, going much beyond the scope of this study. Instead, we opted for a
single parameterization of the snowpack radiative transfer, the 'B60' option from Brun et al. (1992) presented in Lafaysse et al.
(2017), whereby the snow albedo of a layer is a function of its age.

### 2.2.2  Ensemble of meteorological forcings

Meteorological forcings are taken from SAFRAN reanalysis over the Alps and Pyrenees. SAFRAN (Durand et al., 1993) is a
surface meteorological analysis system adjusting backgrounds from NWP model ARPEGE (Courtier et al., 1991) with local
meteorological observations (air temperature, pressure, precipitation, humidity) within so-called massifs of about 1000 km$^2$
(see Fig. 1). SAFRAN analysis is issued separately for each massif in a semi-distributed geometry, i.e within 300 m elevation
bands, aspect and slopes, the main topographic parameters controlling the snow cover evolution. This analysis can be subse-
quently interpolated into specific topographic conditions (i.e. elevation, slope, aspect and local topographic mask) to run snow
cover simulations at specific locations (Vionnet et al., 2016). This means a same analysis is applied to all the points within a





same massif, and interpolated consistently with their topographic parameters, while analyses for neighbouring stations located in distinct massifs will be different.

An ensemble of forcings was generated by applying stochastic perturbations in the same spirit as Charrois et al. (2016) but with slight corrections in the implementation of the perturbations compared with Cluzet et al. (2020, 2021) as described in Deschamps-Berger et al. (in review). The perturbation parameters were taken from Charrois et al. (2016). Precipitation pa-

rameters were adjusted to (i.e. auto correlation time of perturbations $\tau = 1500h$, and dispersion $\sigma = 0.5$) in order to obtain a spread-skill close to 1 for the open-loop run (see Sec. 4.1). We used these perturbed analyses as input for the snowpack simulations at the stations.

### 2.2.3   The Particle Filter in CrocO

The Particle Filter used in this work is based on the version described in Cluzet et al. (2021). We opted for a localised strategy, performing one analysis per simulation point (Farchi and Bocquet, 2018).This approach only considers observations within a disk of a given radius around the considered point. This domain is defined based on the physical understanding of the modelled system. Domain localisation is commonly used in the Ensemble Kalman Filter (EnKF, (Evensen, 1994)) and PF communities to mitigate far-range unrealistic correlations and degeneracy issues (Van Leeuwen, 2009; Poterjoy, 2016; Penny and Miyoshi,

2016; Farchi and Bocquet, 2018) and proved its efficiency in several studies (e.g. Poterjoy and Anderson, 2016; Potthast et al., 2019). Different localisation radius are tested in this study ranging from 17 km to 300 km.

Because PF degeneracy may arise even with a reduced domain sizes, the localisation is completed here by two different strategies described in Cluzet et al. (2021), inflation and k-localisation, leading to the 'rlocal' and 'klocal' algorithms, respectively. Both approaches are iterative, targeting an effective sample size $N_{eff}^*$ if the initial analysis is degenerated ($N_{eff} < N_{eff}^*$). The

rlocal algorithm performs an inflation of observation errors inspired by Larue et al. (2018), while the klocal algorithm selects a subset of observations based on background correlation patterns. It is important to remind that inside a localisation radius, the rlobal method assimilates all available observation stations whereas the klocal method only selects a subset of observations from locations where the ensemble members are sufficiently correlated with the simulation members of the considered point.

**3   Evaluation strategy**

This work aims at assessing the potential transfer of information between points in an HS observation network by means of localized data assimilation, and more specifically to address the questions presented in the end of Sec. 1. To demonstrate that, the data assimilation system must over-perform the state-of-the-art operational deterministic snow cover modelling system from Météo-France (oper) and its ensemble non-operational open-loop counterpart (open-loop).





### 3.1 Setup

Assessing the ability of data assimilation to propagate information requires to use independent data for validation. We opted for a leave-one-out setup in which local observations are removed from the set of observations used in the local PF analysis. Only weekly observations were assimilated, while all available observations between October 1st and June 30th were kept for evaluation.

There are two key design parameters for the data assimilation system: the value of the localisation radius (large or small) and the choice of the PF algorithm (rlocal or klocal). Both exert a direct or indirect control on the number of observations simultaneously assimilated by the PF, and therefore, on its potential degeneracy and its ability to transfer information between locations. Experiments respectively combining the rlocal and klocal algorithm with 4 different localisation radius were conducted: ranging from 17 km, (the radius of an idealised circular SAFRAN massif of 1000 km$^2$) to 300 km (the maximal distance between two observations inside the Pyrenees and the Alps) with two intermediate radius of 35 km and 50 km. The standard deviation of observation errors was set to 0.1 m, as a way to accommodate for measurement and representativeness errors.

Because the klocal approach does not use inflation (except in the case of degeneracy with only one observation), it is quite sensitive to the initial value of observation error. In case of degeneracy, the smaller the observation error, the fewer observations will be selected by the klocal algorithm. For this reason, the klocal algorithm was run with a multiplication factor of 5 on observation error variance (hence a fixed error standard deviation of 0.22 m) , allowing more observations to be assimilated simultaneously.

### 3.2 Evaluation Scores

Several metrics are used in this work to assess the performance of the oper, open-loop and assimilation runs with respect to HS observations. From the ensemble $E_{m,p,t}$ of $N_e$ members $m$ at station $p$ and time $t$, the mean can be computed using Eq. 1:

$$\overline{E}_{p,t} = \frac{1}{N_e} \sum_{m=1}^{N_e} E_{m,p,t} \qquad (1)$$

The mean is a convenient way of synthesizing ensemble properties for evaluation, however, some artifacts can be observed with bounded variables such as HS. On a decaying snow cover for example, the mean will not reach zero until every member has melted. For this reason, the ensemble median $\tilde{E}_{p,t}$ will be preferred in the following. From $\tilde{E}_{p,t}$, we can compute the Absolute Error of the ensemble median compared with the observations $o_{p,t}$ (AE):

$$AE_{p,t} = |\tilde{E}_{p,t} - o_{p,t}| \quad \forall (p,t) \in [1, N_{pts}] \times [1, N_t] \qquad (2)$$

Where $N_t$ is the number of evaluation time steps.

The ensemble bias is defined as the average difference between the ensemble median and the observations (Eq. 3):

$$\text{bias} = \frac{1}{N_t} \frac{1}{N_{pts}} \sum_{t=1}^{N_t} \sum_{p=1}^{N_{pts}} \tilde{E}_{p,t} - o_{p,t} \qquad (3)$$




The Root Mean Squared Error of the median (RMSE) is computed from the AE, following (Eq. 4):

$$\text{RMSE} = \sqrt{\frac{1}{N_t}\frac{1}{N_{pts}}\sum_{t=1}^{N_t}\sum_{p=1}^{N_{pts}} AE_{p,t}^2} \qquad (4)$$

Bias and RMSE can be computed for the oper run (treating it as a single-member ensemble) in order to evaluate the median performance, and can be taken over time and/or space by dropping the time/spatial mean in Eqs.3 and 4. These scores are not sufficient because they reduce an ensemble to its median. The ensemble spread (or dispersion) $\sigma$ (Eq. 5), defined as the average variance, is a first metric to assess an ensemble reliability:

$$\sigma = \sqrt{\frac{1}{N_t}\frac{1}{N_{pts}}\frac{1}{N_e}\sum_{t=1}^{N_t}\sum_{p=1}^{N_{pts}}\sum_{m=1}^{N_e}(E_{m,p,t}-\overline{E}_{p,t})^2} \qquad (5)$$

Reliability is a desirable property for an ensemble, it means that all events are forecast with the right probability regardless the probability value. The pdf of a reliable ensemble matches the actual pdf of observations over a large enough sample. If we introduce the Spread-Skill (SS) as:

$$SS = \frac{\sigma}{\text{RMSE}} \qquad (6)$$

Where sigma must be computed only in the dates and locations where the RMSE is computed. For a reliable ensemble, we have $\sigma \sim \text{RMSE}$ (Fortin et al., 2015), i.e a spread-skill close to unity (necessary but not sufficient condition). This means that the spread is on average a good estimate of the modeling error, which is useful to make decisions. Rank diagrams (Hamill, 2001) are the histogram of the position of the observation within the ensemble and enable to verify the reliability of an ensemble more closely (e.g. Bellier et al., 2017). Their flatness is a stronger condition for an ensemble's reliability than the SS=1.

The Continuous Ranked Probability Score (CRPS, (Eq. 7) Matheson and Winkler, 1976) is an aggregate, ensemble score evaluating the reliability and resolution of an ensemble based on a verification dataset. An ensemble has a good resolution when it is able to issue different forecasts on different events (contrary to the climatology) (Atger, 1999).

If we denote $F_{p,t}$ the Cumulative Distribution Function (CDF) and $O_{p,t}$ the corresponding observation CDF (Heaviside function centered on the truth value), the CRPS is computed at $(p,t)$ following:

$$\text{CRPS}_{p,t} = \int_{\mathbb{R}} (F_{p,t}(x) - O_{p,t}(x))^2 dx \quad \forall (p,t) \in [1, N_{pts}] \times [1, N_t] \qquad (7)$$

The CRPS skill score (CRPSS) is commonly used to compare the performance of an ensemble $E$ to a reference $R$. Despite CRPS can be computed from a deterministic run, $R$ should be preferably an ensemble because comparing CRPS of deterministic and ensemble runs mainly illustrates the obvious fact that an imperfect deterministic run is a poor representation of a probability distribution. The following equation is frequently used:

$$\text{CRPSS*}(E, R) = 1 - \frac{\text{CRPS}(E)}{\text{CRPS}(R)} \qquad (8)$$





| | oper mean (m) | oper RMSE (m) | oper bias (m) | open-loop RMSE (m) | open-loop sigma (m) | open-loop bias (m) |
|---|---|---|---|---|---|---|
| 2009 | 0.28 | 0.27 | −0.02 | 0.28 | 0.28 | −0.04 |
| 2010 | 0.16 | 0.22 | −0.01 | 0.21 | 0.18 | −0.03 |
| 2011 | 0.26 | 0.26 | −0.05 | 0.28 | 0.26 | −0.10 |
| 2012 | 0.44 | 0.37 | −0.03 | 0.39 | 0.38 | −0.11 |
| 2013 | 0.32 | 0.31 | 0.01 | 0.32 | 0.29 | −0.06 |
| 2014 | 0.23 | 0.26 | 0.01 | 0.26 | 0.23 | −0.03 |
| 2015 | 0.24 | 0.27 | 0.01 | 0.27 | 0.25 | −0.01 |
| 2016 | 0.20 | 0.27 | −0.02 | 0.27 | 0.19 | −0.07 |
| 2017 | 0.41 | 0.41 | −0.09 | 0.45 | 0.31 | −0.16 |
| 2018 | 0.23 | 0.31 | −0.07 | 0.33 | 0.19 | −0.12 |

**Table 1.** Yearly performance of the reference runs, in terms of RMSE, bias and spread (sigma).

In this formulation, if E is more skillful than R, CRPSS*(E, R) will be positive, with a perfect score of 1., while less skillful scores range between $-\infty$ and 0, resulting in an asymmetry between positive and negative scores (i.e. CRPSS*$(E,R) =$
$\frac{\text{CRPSS*}(R,E)}{\text{CRPSS*}(R,E)-1}$). We introduce the new formulation:

$$
\begin{cases}
\text{CRPSS}(E,R) = 1 - \frac{\text{CRPS}(E)}{\text{CRPS}(R)} & if \ \text{CRPS}(E) < \text{CRPS}(R) \\
\text{CRPSS}(E,R) = \frac{\text{CRPS}(R)}{\text{CRPS}(E)} - 1 & otherwise
\end{cases}
\tag{9}
$$

With such formulation, CRPS(E,R) $\in [-1,1]$ and CRPS(E,R) = -CRPS(R,E).

## 4 Results

### 4.1 Performance of the reference runs

The operational deterministic run from Météo-France suffers from significant errors (Lafaysse et al., 2013), which we try to reduce by means of assimilation. The open-loop run is a first step to represent modelling uncertainty using an ensemble. Tab. 1 summarizes the yearly performance of both simulations over the 10 years and the 295 stations. Oper and open-loop simulations exhibit almost identical RMSE scores across all years, with an average error of about 0.2-0.3 m. Their RMSE significantly varies (from 0.21 m in 2010 to 0.45 m in 2017 for the open-loop) in proportion with the yearly average snow depth. Oper and open-loop are slightly negatively biased, especially for the open-loop, with a notably strong negative bias during winter 2017 which was exceptionally snowy in the Alps.

Regarding ensemble metrics, the open-loop exhibits Spread-Skills (SS) around 0.9-1 (SS is obtained by dividing the $\sigma$ column by the RMSE column in Tab. 1). SS ranges from a good balance between spread and RMSE in 2009 (SS=1.) to under-





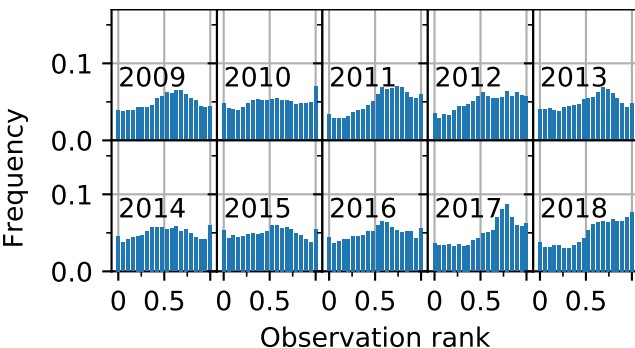

**Figure 4.** Yearly rank diagrams of the open-loop, binned into 20 bins (i.e. for a reliable ensemble, all bars should be on the 0.05 line). Values on the x-axis correspond to the proportion of ensemble members under the observation.

dispersive values (e.g. SS=0.55 in 2018) in the three last years. In Fig. 4, yearly rank diagrams exhibit higher frequencies in their right part, meaning that observations lie preferentially in the upper half of the ensemble, consistently with the biases exhibited in Tab. 1.

A map of the open-loop bias for each station is shown in Fig. 5. The bias is significantly negative in most locations, and its
spatial variability is high, with neighbouring stations exhibiting strong biases of opposite signs, e.g. in the Central Alps. Around Andorra and in the Southern Alps the bias is mostly negative. Some stations exhibit positive biases in the Central Alps, more rarely in the Pyrenees.

## 4.2   Overall results of the assimilation experiments

In this work, we want to compare the performance of the rlocal and klocal algorithm, with different localisation radius (ranging from 17 km to 300 km) with the oper and open-loop runs. Fig. 6 shows the yearly values of RMSE, bias and SS for all these runs. Results show no significant RMSE improvements for the assimilation runs compared with the references. RMSE varies more from one year to another than between assimilation configurations (algorithm and localisation radius). The median RMSE is slightly lower for the intermediate localisation radius of 35 km and 50 km. Compared with the open-loop, assim-
ilation runs significantly reduce the bias both in terms of median value from around -0.06 to about -0.03 and inter-annual variability. Compared with the oper run, the absolute bias of the assimilation runs is higher on average, but in some years, the bias is significantly reduced (e.g. 2015, 2017, 2018).

   In terms of SS, the assimilation runs exhibit values almost twice as small as the open-loop run which has a median value
around 0.85. The SS significantly decreases with an increasing localisation radius both for the rlocal and klocal algorithm. The assimilation strategy without localisation (radius of 300 km) appears as most efficient in reducing biases (lower absolute median, lower inter-annual variability) but yields the lowest spread-skills and highest RMSE of all the assimilation runs sug-

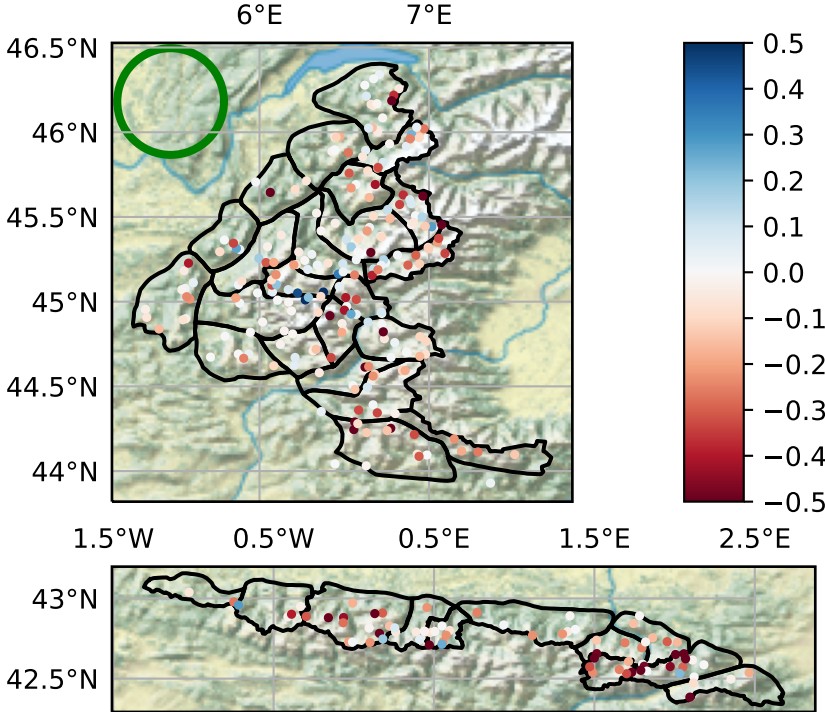

**Figure 5.** Map of the open-loop bias (m) on each station over the ten considered years (same layout as Fig. 1). SAFRAN massifs are outlined in black. The green circle has a radius of approximately 35 km.

gesting that this approach is not the most desirable. The most selective localisation strategies (radius of 17 km) achieve the highest SS, but their inter-annual performance variability is higher than for the other localisation radius.


## 4.3 Factors of variability of the assimilation skill

In the following, we will investigate the different factors of skill variability for the assimilation runs. As described in the previous Sec. 4.2, skill differences between the localised radius of 17-50 km, and between the rlocal and klocal algorithms are small. For the sake of illustration, we decided to focus on the assimilation configuration yielding the lowest median RMSE, the
klocal with a 35 km localisation radius, further referred to as 'klocal' configuration.

### 4.3.1 Spatial variability

Fig. 7 shows boxplots of the daily deviation values (difference between the model median $\tilde{E}_{p,t}$ and the observation $o_{p,t}$) for the klocal and the reference runs grouped per 500 m elevation classes. The bias of the oper varies from slightly positive values





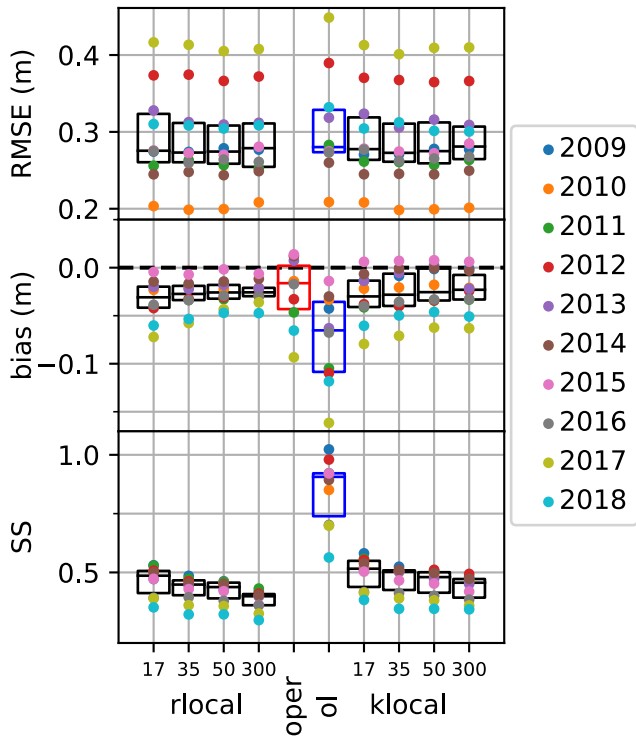

**Figure 6.** Yearly scores of RMSE (top panel), bias (middle) and Spread-Skill (SS, bottom), for the assimilation experiments compared with the oper and open-loop (ol) scores from Tab. 1. On the background are displayed the corresponding boxplots and medians (black bars).

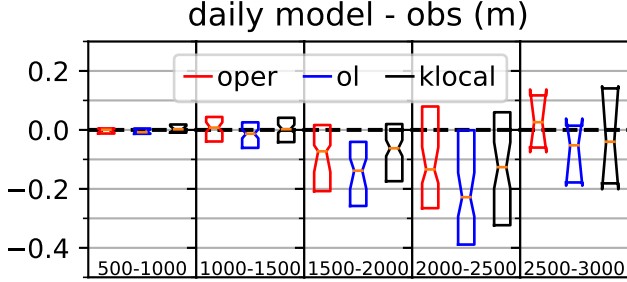

**Figure 7.** Notched boxplots of the daily difference between modelled and observed values (over the 10 years) of the oper (red), open-loop (blue) and klocal:35km (black), by 500 m-wide elevation bands. Occurrences when the three differences are equal to zero are excluded.

between 1000-1500 m to negative values in the range 1500-2500 m to finally a positive bias at the highest elevations. The open-loop exhibits a similar pattern, with a negative shift. The klocal algorithm seems to temper these elevation biases, with lower biases (in absolute value) than the oper both at higher and intermediate elevations.





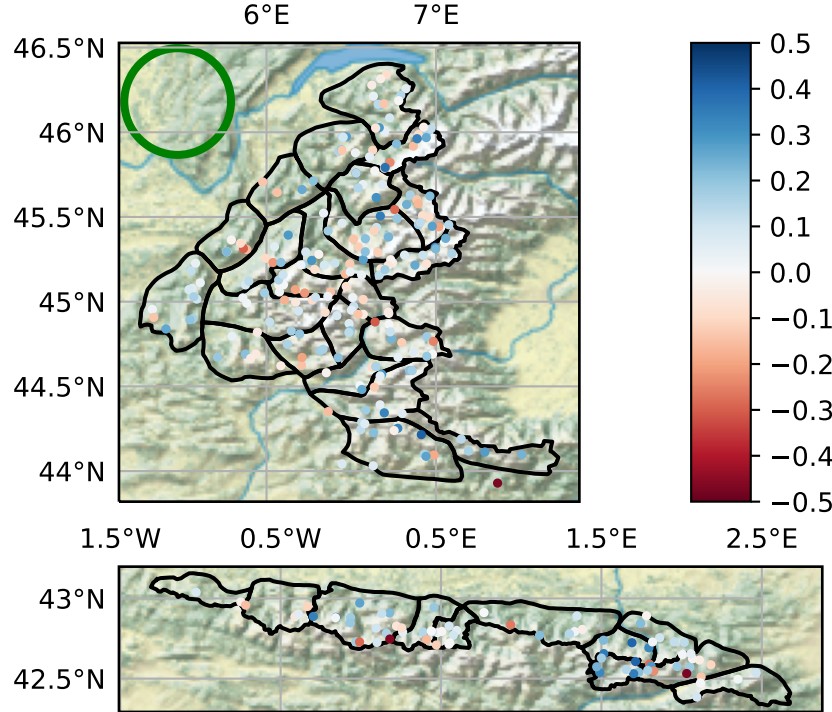

**Figure 8.** Same as Fig. 5, showing the CRPSS of the klocal against the open-loop over the ten years.

Fig. 8 shows the CRPSS of the klocal (using the open-loop as reference) at each station, over the ten years. Overall per-
formance is only slightly positive (blue), but with a non negligible minority of station showing negative CRPSS (red color)
denoting a degradation of performance. Some "clusters" of good performance also appear, as in the Central-Eastern part of the
Pyrenees (Andorra and Haute Ariège) or the Southern Alps, while the performance in the Central Alps and Central Western
Pyrenees seems poor.

Fig. 9a represents the CRPSS as a function of the station elevation. On average, the analysis exhibits positive CRPSS (be-
tween 0. and 0.15) showing that it is more skilful than the open-loop. CRPSS values exhibit a significant spread (of about 0.2)
which results in a number of stations with a degradation of skill by the analysis (negative CRPSS). The average CRPSS varies
with the altitude, increasing from a very low skill (0.-0.03) in the range 1000-1500 m to a significant skill (0.1-0.15) between
1600-2000 m, and finally decreasing to about 0.05 above 2000 m.
Given the strong link between the bias of the open-loop reference and the elevation, the CRPSS was also plotted against the
bias of the open-loop in Fig. 9b. The CRPSS exhibits significant averaged positive values (0.13-0.2) for strong negative biases,
under -0.1. Then, the CRPSS varies from null performance around null bias to significant negative performance for positive





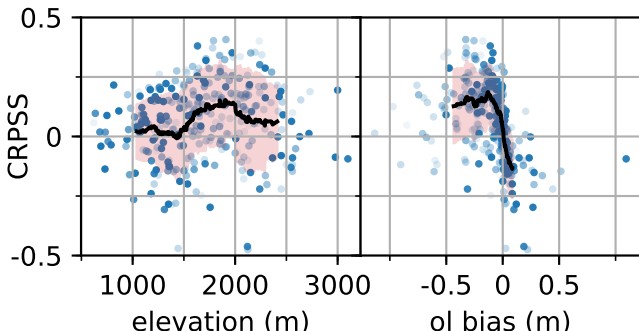

**Figure 9.** Scatter plot of the CRPSS of the klocal run compared with the open-loop for each station over the 10 years, as a function of the station elevation (left panel) and the open-loop bias at each station (right panel). The transparency of the points is related to the proportion of available observations over the validation period. The black line denotes a 51-stations-wide CRPSS rolling average, with an orange shading $\pm 1\sigma$. This average is weighted proportionally to each station transparency.

biases (-0.12).

The density of available observations was pointed as an important factor for the success of the assimilation of in-situ measurements (Winstral et al., 2019; Largeron et al., 2020). We define the observation density as the average number of observations available on each analysis date, divided by the area of the localisation disk. Fig. 10a shows the values of CRPSS as a function of the observation density. CRPSS values are rather spread, and do not seem to vary much with the observation density. On Fig.10 (bottom panel), the open-loop bias is also plotted against the observation density, showing that the highest biases are obtained for the lowest observation densities, although there cannot be any causal relationship as HS observations are not assimilated in the open-loop.

### 4.3.2 Temporal variability

Timeseries of ensemble bias can also provide information on their nature and origin. Fig. 11 shows the timeseries of domain wide ensemble median $\tilde{E}$ against the bias and SS of the several runs in 2009. This year is representative of the different runs behaviours over the 10 years. The bias of the oper run is negative except in April during the melting season. During this year, the bias of the klocal run is centered on zero from mid-January to the end of April. The open-loop is negatively biased for the whole season. Consistently, the ensemble median is the highest for the klocal run. The most interesting feature here, is that the simulation bias is increasing (in absolute value) on several drops, coinciding with increases in $\tilde{E}$ during solid precipitation events (e.g. early December, first week of February, late March, first week of May). The bias difference between the klocal and the open-loop (in mauve) shows the ability of the former to reduce this bias. This reduction is stepwise, with the strongest reductions occurring on analyses (dashed vertical lines) during the accumulation period (e.g. early December, and the two first

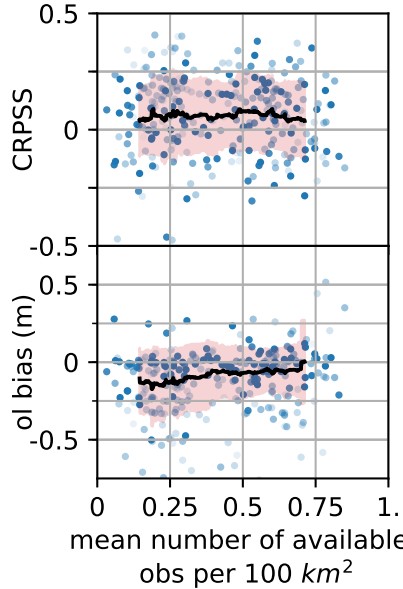

**Figure 10.** CRPSS of the klocal PF as a function of the average density of available observations (top), and open-loop bias as a function of the average density of observations (per $100 km^2$) (bottom)

analyses of January). Between the analyses, and during the melting season, the time evolution of the klocal bias follows the time evolution of the open-loop bias, and the bias difference remains more or less constant.

The SS is an estimate of the ability of ensemble systems to assess their errors (see Sec. 3). Here, consistently with Sec. 4.2 and Fig. 6, we note that throughout the season, the SS of the klocal is inferior to 1 and significantly lower compared to the open-loop. While the SS is similar in both simulations in the early season, klocal analyses seem to coincide with reductions of SS, suggesting that the ensemble spread is more reduced than its error (RMSE) by the PF. In line with the assessment of the reliability, Fig. 12 shows the rank diagrams of the klocal over the 10 years. Compared with the results of the open-loop on Fig.

4, these rank diagrams exhibit a U-shape, consistent with the significant under-dispersion of the klocal. Indeed, by summing the left and right bin frequencies, we observe that the observations lie about 20% of the time in the extremal bins of the rank diagram (twice as much as for a reliable ensemble), and preferentially above, which is consistent with the residual negative bias of the klocal simulation.

## 5    Discussion

In the following, we analyse the strengths and weaknesses of the operational and open-loop simulations and comment on the performance of the data assimilation algorithms in comparison to them.



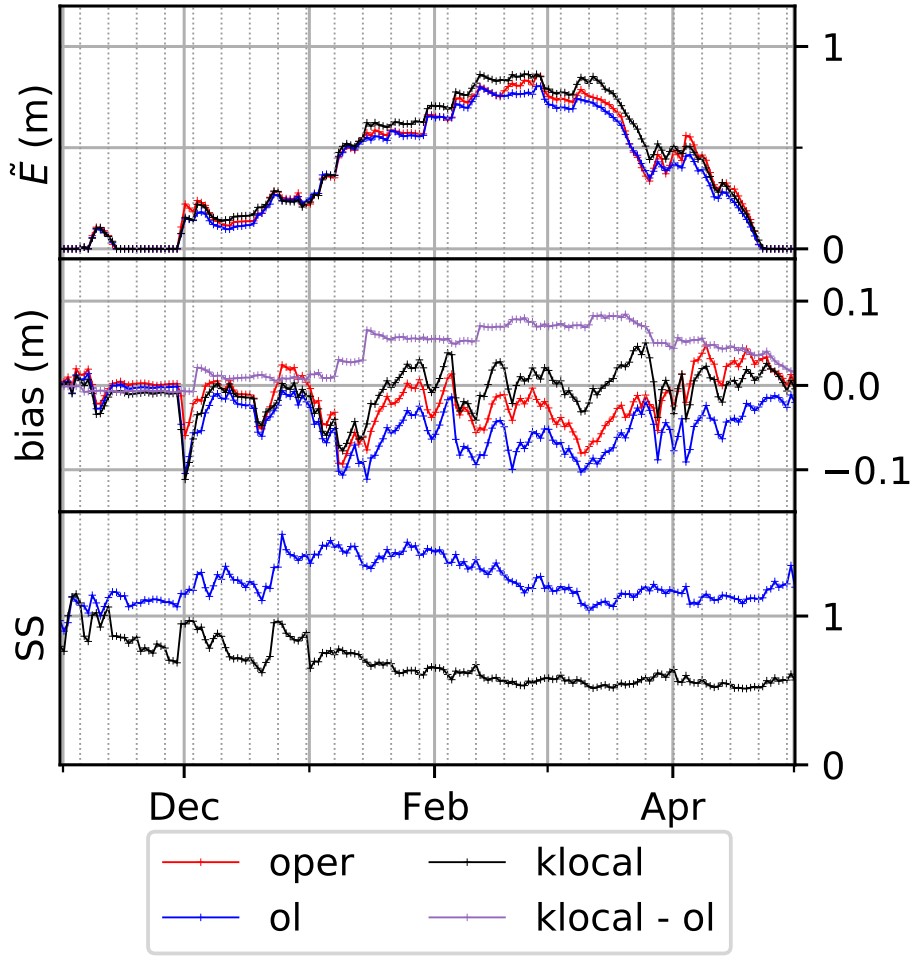

**Figure 11.** Time series of domain averaged ensemble median ($\tilde{E}$) (top), bias (center) and Spread-Skill (SS, bottom) for the winter season 2009-2010, for the oper (red), open-loop (blue) and klocal (black). The bias difference between the klocal and the oper is also plotted in mauve in the middle panel. Vertical dashed lines correspond to the assimilation dates. The onset (October) and late season (June-July) are not plotted for the sake of clarity.

## 5.1 On the performance of the reference simulations

The performance of the operational simulation has been regularly assessed until recently (Durand et al., 2009a; Vernay et al., in prep). Overall, it is an accurate modelling system whose potential has been demonstrated in several recent climate studies and projections (e.g. López-Moreno et al., 2020; Verfaillie et al., 2018). However, it exhibits a contrasted regional performance (Vernay et al., in prep, Fig. 13), and its errors are badly known at high altitude, due to the lack of observations (Fig. 12 of Vernay et al. (in prep)). This is a common issue in mountainous areas (Frei and Schär, 1998) and is detrimental for the use of the operational chain for all applications (e.g. avalanche hazard forecasting, hydrology etc.).





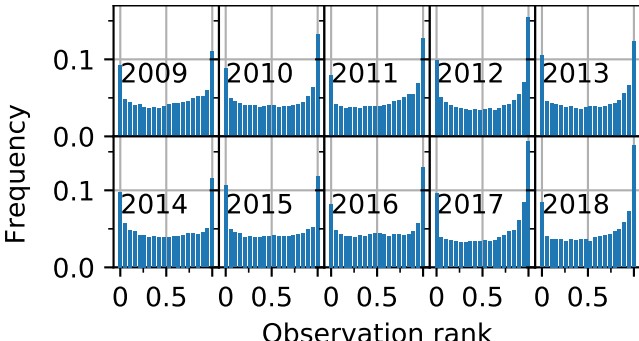

**Figure 12.** same as Fig. 4 for the klocal.

Results from Tab. 1 shows that the operational version of the system, and its ensemble version, the open-loop, have comparable
RMSE. The open-loop run enables to satisfactorily account for modelling uncertainties and errors, since its SS is slightly below
unity over the ten years. This means that on average, the ensemble spread is almost a reliable estimate of the modelling error.
This feature could be valuable for forecasters (Buizza, 2008).

Tab. 1, and Figs. 6 and 11 show that the open-loop is negatively biased compared to the oper. This could be due to the cen-
tered stochastic perturbations (Charrois et al., 2016; Deschamps-Berger et al., in review), or a bias in the ESCROC multiphysics
model configurations (Lafaysse et al., 2017). However, the oper model configuration is not expected to be perfectly centered in
the open-loop, as several configurations, such as the parametrization of surface heat fluxes, ground heat capacity or fresh snow
density strongly influence the resulting modelled snow depth. Strong increases in the oper and open-loop biases match with
precipitation events, and they are only partly compensated by the following snow settling period (see Sec. 4.3.2), suggesting
that it is likely that error compensations take place in the oper chain, between solid precipitation amounts, fresh snow density,
snow compaction, and ablation processes as suggested by results from Quéno et al. (2016). Evaluation with co-located SWE
and HS data would help disentangle this situation (e.g. Smyth et al., 2019).

Biases of the oper and open-loop strongly depend on the altitude (Fig. 7) in a pattern that matches the evaluation from Vernay
et al. (in prep), though on a smaller number of stations and considered years. They are unambiguously negative in the range
1500-2500 m, and more variable above, probably due to a higher snow cover variability, and depending on the considered
region. The most likely cause for strong negative biases at those altitudes is an underestimation of solid precipitation amounts
in gauges (Kochendorfer et al., 2017), and consequently in SAFRAN, as evidenced by (Quéno et al., 2016) during strong
precipitation events.



## 5.2 On the PF strategies

In general one of the primary motivations of the domain localisation is to prevent the PF from degenerating (Farchi and Bocquet, 2018). In our case, as evidenced by the reasonable performance of the rlocal with a 300 km localisation radius (e.g. therefore simultaneously assimilating up to 217 observations in the Alps), domain localisation is not required against PF degeneracy thanks to the mitigations (i.e. inflation or k-localisation) developed in Cluzet et al. (2021). Here, localisation is rather used to adapt to the structures of errors of the reference run. From Fig. 5, it seems that open-loop bias is systematic and widespread. Then a large localisation radius, averaging a significant number of observations, seems a good option. However, we also see regional structures in this bias, probably inherited from the oper (Vernay et al., in prep). They are likely due to the fact that SAFRAN analyses are performed at the scale of the massif. To address this type of error, reducing the localisation radius is probably a better option. Finally, errors structures can depend on other parameters such as the elevation, and vary in time. In this situation, the klocal approach might be more adapted, since it adjusts the observation selection on the model background correlation patterns. However, these background correlation patterns could sometimes be unrealistic, and therefore, misleading for the algorithm.

The klocal algorithm, by construction, selects observations from locations that are correlated in the model's point of view. However, because we apply spatially homogeneous perturbations to the meteorological forcings, strong large scale background correlation patterns are present in the open-loop, even between the Alps and Pyrenees (not shown). These strong, potentially artificial, large scale correlation patterns could hamper the performance of the klocal PF, leading it to assimilate very distant observation with no actual link with the considered location. Conversely, a completely random field of perturbations would prevent the algorithm from propagating any information between locations (Magnusson et al., 2014; Cantet et al., 2019). Using physically-based meteorological ensemble, such as PEARP (Descamps et al., 2015), used in Vernay et al. (2015) or AROME-EPS (Bouttier et al., 2016), or spatially correlated perturbation fields (Magnusson et al., 2014), could lead to more realistic correlation fields, but this goes much beyond the scope of this study, as actually, domain localisation prevents the klocal from assimilating too distant observations.

## 5.3 Overall performance of the assimilation compared with the references

Here, we discuss the ability of the proposed assimilation approaches (with several localisation radius) to succeed in reducing the modelling errors from the oper and open-loop shown in Sec. 5.1. Aggregated results from Fig. 6 show that none of the proposed assimilation configurations enable us to significantly reduce overall modelling errors compared to the operational run. However, they overcome the significant negative bias of the open-loop they originate from, but at the expense of a strongly under-dispersive spread-skill. The bias reduction seems more efficient and stable (i.e. less variable from year to year) with the rlocal than with the klocal, and with a larger localisation radius, which makes sense as the open-loop bias is widespread (e.g. Fig. 11) and both tend towards assimilating more observations at the same time. However, the RMSE is slightly larger for the





largest localisation radius, and the spread-skill is strongly reduced too.

There are two reasons why the operational run could not be beaten by the assimilation in terms of RMSE. First, its error

may be of a same magnitude than the natural variability of point scale observations and in that case, no added value can be extracted even from nearby observations, or similarly, there are too few observations to efficiently constrain modelling errors. Increasing the observation density could be an option to overcome this issue. However, our results do not show a strong relationship between assimilation skill and density (Fig. 10, see Sec. 5.5 later on). Another explanation could be that there still remain systematic errors to correct, namely biases (as suggested by Fig. 7) but it is difficult to propagate information between

locations. In an idealised case, (Cluzet et al., 2021) showed that the potential to propagate information from HS observations across elevations is limited. Here, modelling errors are not systematic and strongly vary with the altitude (Fig. 7). If the ensemble does not account for this specific bias structure, an observation at an elevation affected by a positive bias could never help choose the best member configuration for an elevation affected by a negative bias.

### 400    5.4    On the difficulties faced by assimilation algorithms

In this part, we comment the performance of the klocal with a localisation radius of 35 km assimilation configuration against the open-loop. Despite it does not outperform other configurations significantly, the klocal seems best suited to solve the bias-elevation relation in the references and an intermediate localisation radius enables to adapt to local error structures (see Sec. 5.2).

The CRPS improvement is the highest for intermediate elevations coinciding with the highest open-loop negative bias (Fig. 9, the latter being consistent with Cluzet et al. (2021) who showed that the largest improvements were obtained in the presence of systematic biases.

However, the klocal is strongly underdispersive, contrary to the open-loop which achieves a SS around 1, and therefore is significantly less reliable as evidenced by the U-shaped rank diagrams in Fig. 12. As the CRPS is a measure of both accuracy

and reliability, it seems surprising to see that the klocal is more skillful than the open-loop in terms of CRPS, with average positive CRPSS around 0.06 (Fig. 9).

This under-dispersion is not satisfactory because it implies that the assimilation run is too confident about its simulated distributions. This is a general issue for all the presented assimilation strategies (Fig. 6). In additional experiments (not shown), the assimilation frequency was reduced to 14 days, in order to let the ensemble spread increase between assimilation dates. It

seems a reasonable value according to e.g. Smyth et al. (2020) and Viallon-Galinier et al. (2020), and resulted in an increased spread, but was detrimental to the RMSE. We did not consider increasing the target efficient sample size, $N^{*}_{eff}$, which is set to 100. This value, is much higher than previous studies (Larue et al., 2018; Cluzet et al., 2021) and was chosen as preliminary experiments (not shown) with values of 25 and 50 which gave an even lower SS. Finally, the spread of the stochastic perturbations on the forcings could be increased, or statistically calibrated distributions of the main forcing variables (e.g. Taillardat

and Mestre, 2020) could be used.





Nevertheless, obtaining a perfect spread-skill may be a challenging goal for our assimilation system. Indeed, under dispersion is a common issue in the NWP (e.g. Bellier et al., 2017) and snow cover modelling communities (Lafaysse et al., 2017; Nousu et al., 2019). Then, the ensemble modelling chain does not account for two important processes affecting the observations at the stations: the variability of the meteorological conditions inside SAFRAN massifs, and the snow redistribution by wind

(Vionnet et al., 2018; Mott et al., 2018).

Data assimilation is also known to partly compensate for these phenomena via error compensation (Klinker and Sardeshmukh, 1992; Rodwell and Palmer, 2007; Wong et al., 2020). For example, an ablation event in one observation can be compensated in the Particle Filter by selecting some members with a lower precipitation factor or a compaction scheme with a higher settling. This compensation immediately results in lower errors, but implicitly, the model does a wrong assumption, which results in

being over confident, thus with a lower spread. The only way to mitigate for this over confidence is to account for any relevant physical phenomenon, which is a desirable goal of course, but a real challenge when it comes to snow transportation by wind and local meteorology. This goal is to date out of reach at the temporal and spatial scale of this study.

Despite these limitations, the assimilation shows some ability to correct weaknesses of the reference runs. The first one is

the significant bias above 1500 m in the reference run (Fig. 7). This bias probably originates from a lack of meteorological observations in SAFRAN analysis at those altitudes (see Sec. 5.1 and Fig. 4 of (Vernay et al., in prep)). In the range 1500-2000 m, the klocal has a significantly lower bias than the open-loop. There is a lower benefit at higher elevations, above 2000 m. (Fig. 9), maybe owing to the fact that snow cover variability is higher, in particular due to stronger winds. There are also less observations available, and a less clear bias at this altitude (there seems to be a transition from a negative bias to a positive bias),

reducing the odds of a successful assimilation. Unfortunately, such elevations are key for avalanche activity (Eckert et al., 2013; Lavigne et al., 2015). Another good feature of the assimilation is to improve the accuracy in areas where the references are less accurate due to a lack of meteorological observations, namely Andorra and Haute-Ariège in the Pyrenees, and Ubaye, Haut Verdon and Mercantour in the southern Alps (Fig. 8). Both features underline the complementarity between HS observations and the meteorological observations already assimilated in SAFRAN.


### 5.5 Performance in relation to the density of observations

The density of in-situ observations has been pointed out as a critical parameter for the success of data assimilation (Largeron et al., 2020). Winstral et al. (2019) managed to strongly reduce modelling errors with a high observation density, (about 1 observation site every 100 $km^2$). Because of natural variability, they considered detection of systematic errors may be more

difficult with a lower density. Our study case explores a wide range of observation density (Fig. 10), from about 0.1 to 0.8 observations every 100 $km^2$ (accounting for the availability of observations). Yet, as mentioned in Secs. 4.2 and 5.1, the assimilation performance relative to the open-loop does not decrease with a lower observation density. It may be due to the fact that the assimilation is efficient only for strong open-loop negative biases (Fig. 9b), which seems the highest where the station density is the lowest (Fig. 10b). In other words: the assimilation can not beat the open-loop in the most densely observed areas





(e.g. in the Northern Alps, where the observation density is similar to the studies of Magnusson et al. (2014) and Winstral et al. (2019)) because the open-loop performance is already high there. This behaviour is surprising as the open-loop is independent from these HS observations. However, (Vernay et al., in prep) showed that the best results of the oper reanalysis are obtained the Northern Alps. Indeed, it is likely that HS observation density and the density of the weather stations used by SAFRAN to analyse the meteorological forcings are correlated. Both (at the exception of the Nivôse and EDF nivo stations for the HS

observations) are actually related to human implantation in the valleys and the presence of ski resorts. A higher weather station density for SAFRAN is likely to result in more accurate meteorological forcings, thus reducing the bias of the reference runs, which finally leaves less room for improvement by the assimilation.

This assumption may guide the strategies of definition of snow cover networks, not only in terms of observation density but also in terms of localisation. Our study suggests that snowpack observations do not yield significant improvements in areas

where a sufficient amount of meteorological observations is already assimilated in the snowpack modelling chain (here, in SAFRAN). The assimilation of snow depth observations rather gives significant improvements at higher altitudes, and in areas where model errors are larger, generally corresponding to areas where less meteorological observations are assimilated. This result could be verified in future work, in either semi distributed or distributed frameworks, validated by e.g. satellite retrievals of the snow cover fraction (Magnusson et al., 2014).


### 5.6 Towards the assimilation in a semi-distributed geometry?

The aim of this study was to assess the potential of the assimilation of in-situ HS observations to correct nearby simulations, in view of applying it in a semi-distributed or distributed framework (Cluzet et al., 2021), in a similar strategy as Magnusson et al. (2014) and Griessinger et al. (2019). We used CrocO (Cluzet et al., 2021), an ensemble system accounting for meteorological

and snowpack modelling uncertainties, using a Particle Filter to assimilate spatialised snowpack observations.

The results are mitigated: an added value is observed only when initial modelling errors are large (Fig. 9b), similarly to results obtained by Winstral et al. (2019). In the Northern Alps, Western Pyrenees and under 1500 m, the added value is null on average, and seems too insufficient to be of a real use. Over these areas, it seems that there is no room for improvement with data assimilation of point scale HS only. There, simulation accuracy may be more limited by snow related processes such as wind

drift and uncertain physical processes resulting in snow cover variability, than by meteorological errors. The use of spatialised satellite retrievals (Margulis et al., 2019; Cluzet et al., 2020) to better constrain snow cover variability, or a finer correction of meteorological forcings using radar precipitation data (e.g. Birman et al., 2017; Le Bastard et al., 2019) in combination with higher resolution NWP models and their ensemble counterparts, might be a solution.





## 6 Conclusions

This study investigates the potential for localised versions of the Particle Filter to spatially propagate information from in-situ observations of the height of snow (HS) in an ensemble of snowpack simulations. Compared with state-of-the-art deterministic and ensemble open-loop approaches, over ten years, we demonstrate that substantial improvements are only obtained in locations and elevation ranges where the reference errors are the highest. These areas correspond to locations where the density of meteorological observations, which are crucial for the correction of the meteorological forcings within SAFRAN analysis scheme, is the lowest. This demonstrates a good complementarity with the meteorological observation analysed by SAFRAN to reduce the current errors of the operational chain.

Previous studies already demonstrated the added value of in-situ HS observations in a similar setting with a dense observation coverage (Magnusson et al., 2014; Winstral et al., 2019). It was suspected that lower observation densities would reduce the potential for assimilation. Here, we exploit data with a wide range of densities, generally lower than these studies, and find no sensitivity of the assimilation performance to the observation density. This finding may be specific to the error structures of the reference simulations, which are correlated with the observation density.

Results also show that intermediate localisation strategies between 35-50 km of radius yielded slightly lower errors than a strategy addressing large scale errors only (300 km), while lower radius (17 km) may be too small to capture the snow cover variability where the density of observations is too small.

Our results finally show a good complementarity between the HS observations and meteorological observations already assimilated in the modelling chain, in particular in the most remote areas. This result is encouraging in the way of reducing the weaknesses of the current operational modelling chain, and shows that even scarce in-situ snowpack observations could be beneficial for snow cover modelling over large areas.

*Code availability.* The Crocus snowpack model (including all physical options of the ESCROC system) and the Particle Filter algorithm are developed inside the opensource SURFEX project. The source files of SURFEX code are provided at 10.5281/zenodo.5111449 to guarantee the permanent reproducibility of results. However, we recommend potential future users and developers to access to the code from its git repository (git.umr-cnrm.fr/git/Surfex_Git2.git, tag CrocO_v1.1). Experiments were pre/post-processed using CrocO_toolbox package. It is available on Github (https://github.com/bertrandcz/CrocO_toolbox, release v1.1) along with a documentation.

However, this software could not be applied outside Météo-France HPC environment, CrocO python software offers the possibility to run CrocO simulations locally. This functionality was not used here due to the high numerical cost of our simulations, which required the use of Météo-France HPC environment.

*Data availability.* SAFRAN reanalyses, corresponding to the unperturbed forcings, are available at: http://dx.doi.org/10.25326/37#v2019 (need to select the *postes* domain). Additional input data necessary to reproduce the manuscript simulations and figures are provided at 10.5281/zenodo.5115557 . This archive includes : namelists, configuration files, spinup files and pre/post-processing scripts. Simulation



outputs represent a considerable amount of data (300+Go) and are archived in Météo-France archiving system and can be accessed upon reasonable request to Matthieu Lafaysse (matthieu.lafaysse@meteo.fr).

*Author contributions.* BC, MD and ML conceived the study, BC performed the simulations, treated the observations and wrote the manuscript, ML downloaded the observations, CDB updated the routines to generate the forcings and helped with the interpretation of CRPS, MV performed the oper runs. All authors contributed to the analysis of results

*Competing interests.* The authors have no competing interests to declare.

*Acknowledgements.* XXXXXXXXX



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
