# Peer review of "Propagating information from snow observations with CrocO ensemble data assimilation system: a 10-years case study over a snow depth observation network"

_The Cryosphere, 2021_

## Author Comment (AC1)

**RC1**: 'Comment on tc-2021-225', Anonymous Referee #1, 21 Sep 2021

The authors would like to thank the reviewer for this throughout review, and in particular for the question of the co-located meteorological and snow observations. Several adjustments were made to the paper in line with this, and a new figure was added. Note that some slight changes were also made in the manuscript in order to improve its clarity, and are visible in the track changes. In the following, the reviewer initial comments are written in black, our answer in blue and the corrections in the paper are highlighted in red.

**Summary of the paper**

Cluzet et al. present an evaluation of the CrocO snow data assimilation system which is based on a Particle Filter technique that propagates information spatially from areas with observations (e.g., height of snow, HS) to unobserved areas. Using leave-one-out validation, the performance of this system is compared to that of the ensemble (i.e., open-loop) and operational model counterpart at hundreds of snow depth stations in the French Alps, Pyrenees, and Andorra over 10 years. Different localization schemes (rlocal vs. klocal) and radii are tested, and showed only minor changes in skill. All Particle Filter configurations showed improvements relative to the open-loop but no clear improvement over the operational model. Using the klocal scheme at 35 km for the rest of the study, the authors show it tends to mitigate the negative bias in the open-loop at certain elevations (1500-2500 m) and locations (e.g., Central-East Pyrenees). The klocal scheme shows the highest skill at locations where the open-loop has negative bias, with reduced skill when the open-loop has positive bias. The skill of the klocal PF was found to be insensitive to the density of HS observations, but the magnitude of open-loop bias decreased with increasing HS observations (a peculiar result). The study suggests there is complementarity in that areas with high errors in meteorological data (due to low station data) may be mitigated by a Particle filter with spatial propagation of snow information.

**Recommendation**

Overall, I find that this paper is of good quality and provides additional application and evaluation of the CrocO Particle Filter developed in recent work by the authors. It is conducted at a more coarse model scale (and less dense observation network) than previous studies (e.g., Magnusson et al 2014; Winstral et al., 2019) and is hence a unique, complementary contribution. I think there are some aspects that could be strengthened in the discussion and possibly the presentation of results, depending on what information is available about the HS stations. The issue is that the stations may benefit both the open-loop and oper (through providing meteorological data to the reanalysis) as well as the PF (through providing HS observations for assimilation). Disentangling these competing contributions to the model configurations could be useful in a revised manuscript.

**Comments**

The authors would like to thank the reviewer for this very acute remark. We completely agree that these observations may have an impact on the performance of the reference runs (openloop and oper), thereby reducing the potential for improvements for the PF assimilating snow observations.

Over the considered period of time, 438 observation sites provided precipitation observations to SAFRAN over this area in the snow season (November-April). These stations are mostly located at lower elevations (below 1500 m) as presented in Fig.4 of Vernay et al. (in review). Among them, 164 stations correspond to locations with snow depth measurements included in the present study. XXX other stations provide snow depth measurements but without precipitation data. We reproduced below Fig. 10 of the submitted manuscript, highlighting in green the 79 stations having an almost-daily precipitation observation frequency (150 obs on average between November and April), (see Fig. 1 here). This figure shows that the performance of the assimilation is similar in the locations with a lot of precipitation observations than in the others.

[Figure]

*Illustration 1: R1.1: like Fig. 10, with green dots corresponding to the 79 stations with more than 1500 precipitation observations. The grey line is the average of these stations, and the black line is the average of the rest of the stations.*

The reason is that SAFRAN does not assimilate precipitation observations locally, but performs a single meteorological analysis within each massif (see l. 128-136 of the submitted manuscript). Precipitation observations may therefore impact the reference runs at neighboring locations, and coincidentally the performance of the reference runs might increase with the density of the available precipitation observations.

This is the hypothesis that we formulated on l. 458-459: the increase in performance of the reference runs with the snow observations density should be explained by the fact that the snow observation density is correlated to the meteorological observation density. To confirm this, we performed a more thorough analysis of the relationship between the spatial density of snow observations and the sparial density of precipitation observations assimilated by SAFRAN. The outcome is presented in Fig. 12 of the revised manuscript, and confirms our statement l. 458-459: the density of snow observations increases with the density of precipitation observations used by SAFRAN, both in the Alps and Pyrenees.

Several changes were performed to reflect this and answer the reviewer's remark in a concise way:

a. more details were included on the precipitation observations, the following sentences being added in Sec. 2.2.2 (see track-changes):

Over the considered period of time, 438 observation sites provided precipitation observations to SAFRAN between November and April. These stations are mostly located at lower elevations (below 1500 m) as presented in Fig.4 of \citet{vernay2021meteorological}. Among them, 164 of these sites correspond to locations with snow depth observations included in the present study.

b. Fig. 12 was added and Sec. 5.5 amended to confirm our hypothesis on the correlation between the HS and precipitation observation densities.

The density of in-situ observations has been pointed out as a critical parameter for the success of data assimilation \citep{largeron2020towards}. \citet{winstral2019bias} managed to strongly reduce modelling errors with a high observation density, (about 1 observation site every 100 \unit{km^2}). Because of natural variability, they considered detection of systematic errors may be more difficult with a lower density. Our study case explores a wide range of observation density (Fig. \ref{fig:res_CRPSS_bias_density_scatter}), from about 0.1 to 0.8 observations every 100 \unit{km^2} (accounting for the availability of observations). Yet, as mentioned in Secs. \ref{sec:res_assim} and \ref{sec:disc_ref_perf}, the assimilation performance relative to the open-loop does not decrease with a lower observation density. It may be due to the fact that the assimilation is efficient only for strong open-loop negative biases (Fig. \ref{fig:res_CRPSS_scatter}b), which seems the highest where the station density is the lowest (Fig. \ref{fig:res_CRPSS_bias_density_scatter}b). In other words: the assimilation can not outperform the open-loop in the most densely observed areas (e.g. in the Northern Alps, where the observation density is similar to the studies of \citet{magnusson2014assimilation}

and \citet{winstral2019bias}) because the open-loop performance is already high there. This behaviour is explained by the fact that the HS observation density is correlated with the density of precipitation observations used by SAFRAN to analyse the meteorological forcings (see Fig. \ref{fig:corr_HS_precip} and Sec. \ref{sec:forc})). Both (at the exception of the Nivôse and EDF nivo stations for the HS observations) are actually related to human implantation in the valleys and the presence of ski resorts. A higher weather station density for SAFRAN is likely to result in more accurate meteorological forcings, thus reducing the bias of the reference runs, which finally leaves less room for improvement by the assimilation.\\

1.A result highlighted in the paper is that the open loop has lower performance in areas with fewer stations and decreased bias as station density increases (L. 299-301 and Fig. 10), but this seems coincidental, as the open loop does not integrate HS observations. Is this more suggestive of reduced quality in the reanalysis forcing data, given the lower availability of surface stations? Alternatively, I wonder how many of the HS stations include meteorological observations that are integrated into the reanalysis data (see lines 92-93)? Overall, the authors should carefully consider the two sources of data offered by stations (meteorology and HS). It might seem that a lack of stations would influence both the open loop and DA for that reason. The authors seem to understand this later in the paper (L. 458-459), so a consistent viewpoint should be included through the paper. Is it possible to identify which HS stations have met data that are being integrated into the reanalysis forcing data? If so, it may be necessary to differentiate the results based on whether the HS stations have met data that are contributing to the forcing data or not.
This point is answered above.

2.The leave-one-out validation (L. 63-65, 166-169) should be clarified whether this is done spatially (e.g., one station is removed) or temporally (e.g., individual HS data are removed at a station). I am still unsure after re-reading these parts.
The purpose of the leave-one out approach is to assess the propagation of information in a localised setting. In this approach, any available local observation is excluded from the assimilation, so that we can evaluate what is the impact of nearby observations on this location.l. 63-65 were amended to clarify this point:

To assess the potential transfer of information, we opt for a leave-one-out approach \citep[e.g.][]{slater2006snow}, whereby the assimilation is performed considering neighbouring observations, but discarding any local observation. The assimilation performance can be then evaluated independently from these local observations. If such potential transfer could be demonstrated, ...

3.The study highlights a range of elevations (1500-2500 m) where is a strong negative bias in the open loop and oper simulations. The role of gauge undercatch is discussed as the most likely cause of this bias (L. 351-353). However, I do not find this convincing, as gauge undercatch is likely a universal problem for measuring precipitation and snowfall in mountain environments. Is there another factor that might explain the bias in this specific elevation zone? Are the wind speeds higher, and hence larger undercatch errors at these elevations? Or is this suggestive of an oddity in the HS station data, for example, more ski area observations in these zones, which may be biased toward higher snow depths?
Here, higher winds at higher elevations was implicitly the factor behind higher under-catch at higher elevations. To our knowledge, the driving factor for the localisation of ski resorts is the general exposition (preferentially north), elevation,and economic potential, and not potential local higher snow amounts (there are also resorts in snow scarce areas). And observation sites are flat and away from any human influence such as grooming of snow-making. The corresponding sentence l 351-353 was amended to reflect that higher winds may be the driving factor.
...and depending on the considered region. The strong negative biases at those altitudes may be explained by higher wind speeds, causing an underestimation of solid precipitation amounts in gauges \citep{kochendorfer2017quantification}, and consequently in SAFRAN, as evidenced by \citep{queno2016snowpack} during strong precipitation events.

4.In the discussion, the authors discuss variability of snow redistribution by wind in the context of error compensation with the DA system (L. 423-432). One aspect that is missing here is the impact of scale – both the process scale and the model scale – and this is something that should be identified and discussed. The resolution of the model in this study is quite large, such that a process like wind redistribution cannot be represented explicitly, and ultimately the variability must be handled in other manners (e.g., as a sub-grid parameterization, see Clark et al., 2011 for example; or improving the model resolution to a finer scale). Please comment more on this issue.

This point is quite technical. Indeed, as stated in l 132-134, the meteorological analyses are determined by topographic classes at the scale of SAFRAN massifs, but are then downscaled to the topographic conditions and terrain shading of the stations where the simulations are performed. They are conceptually very close to point simulations forced with local meteorological observations, so that observations and simulations are the representation are directly identifiable. However, and as pointed out by the reviewer, our aim in l. 423-432 was to stress out that unrepresented processes in the model (mainly wind drift and intra-massif variability), and the limited representativeness of snow observations at the plot scale may explain discrepancies between modelled and observed variables, and induce error compensation by the assimilation algorithm. The corresponding paragraph from Sec 5.4 was rewritten to reflect this:

Nevertheless, obtaining a perfect spread-skill may be a challenging goal for our assimilation system. Indeed, under dispersion is a common issue in the NWP \citep[e.g.][]{bellier2017sample} and snow cover modelling communities \citep{lafaysse2017multiphysical,nousu2019statistical}, and can be explained here by several factors. On the one hand, despite meteorological forcings are downscaled to the stations (see Sec. \ref{sec:forc}, so that there is no scale mismatch), the ensemble modelling chain does not account for two important processes affecting the observations. The variability of the meteorological conditions inside SAFRAN massifs is limited to topographic parameters (including local masks) so that two distant stations with the same topography will receive the exact same forcing, and the snow redistribution by wind is not represented \citep{vionnet2018,mott2018seasonal}. On the other hand, the representativeness of observations is limited by plot-scale variability.
\\
Data assimilation is known to partly compensate for such mismatches via error compensation \citep{klinker1992diagnosis,rodwell2007using,wong2020model}. For example, an ablation event in one observation can be compensated in the Particle Filter by selecting some members with a lower precipitation factor or a compaction scheme with a higher settling \citep{deschamps2021assimilation}. This compensation immediately results in lower errors, but implicitly, the model does a wrong assumption, which results in being over confident, thus with a lower spread. The only way to mitigate for this over confidence is to account for any relevant physical phenomenon, which is a desirable goal, but a real challenge when it comes to snowdrift by wind, local meteorology and plot-scale variability. This goal is to date out of reach at the temporal and spatial scale of this study.\\

We also clarified the model configuration by stating more clearly that SAFRAN analyses are downscaled at the stations in Sec. 2.2.2:

Meteorological forcings are taken from SAFRAN reanalysis over the Alps and Pyrenees. SAFRAN \citep{durand1993} is a surface meteorological analysis system adjusting backgrounds from NWP model ARPEGE \citep{courtier1991} with local meteorological observations (air temperature, pressure, precipitation, humidity) within so-called massifs of about 1000 \unit{km^2} (see Fig. \ref{fig:meth_obs_density_massif}) and further downscaled to the stations of our study. [...] SAFRAN analysis is issued separately for each massif in a semi-distributed geometry, i.e within 300 \unit{m} elevation bands, aspect and slopes, the main topographic parameters controlling the snow cover evolution. This analysis is subsequently downscaled into the specific topographic conditions (i.e. elevation, slope, aspect and local topographic mask) of the simulated station \citep{vionnet2016}.

**General comments**

- The argument that weather stations provide better estimates of surface meteorology than NWP may not be a widely supported viewpoint, for instance see Lundquist et al. 2019. Please comment.

This comment refers to an introduction paragraph l. 30-37. We agree that the sentence 'Information from weather stations located in the mountains provide better estimates of surface meteorology than Numerical Weather Prediction (NWP) models' is wrong, particularly at the light of Lundquist et al., (2019). In particular, NWP models are much more able to capture the spatial variability of meteorological conditions than in-situ observation networks. We want to reformulate this sentence because our point is rather to mention the complementarity between these sources of information:

In that context, additional sources of information are needed to mitigate snowpack modelling uncertainty in the mountains. Observations from weather stations located in the mountains can be used to correct Numerical Weather Prediction (NWP) model outputs. Dedicated downscaling and analysis schemes such as SAFRAN \citep{durand1993} or RhiresD interpolation in Switzerland \citep{frei1998precipitation} can be used to efficiently reduce the large errors of the NWP models in the mountains, in particular by the assimilation of local precipitation observations.

- There are many instances where papers that are "in prep" or "in review" are cited, for instance Vernay et al., Deschamps-Berger et al., etc. Please check with the TC journal guidelines on whether these types of papers (not yet published) are permitted to be cited.

We checked on this webpage: https://www.the-cryosphere.net/submission.html#references. The "in prep"and "in review" papers are now "in review" or published.

**Technical corrections**

- Line 13: Add "of" after "strategy". OK

- Lines 53: Add "the" before "Météo-France". OK

- Line 71: Replace "Does" with "Can". Also, delete "manage to". OK

- Line 84: Add "Alps" after "Northern". OK

- Line 104: Replace "less" with "fewer". OK

- Line 137-142: Can you include a brief summary of the key elements of the ensemble generation technique? This would be useful for someone who has not read Cluzet et al. (2020, 2021) or who needs a reminder about the methods.

An ensemble of forcings was generated by applying stochastic perturbations in the same spirit as \citet{charrois2016} but with slight corrections in the implementation of the perturbations compared with \citet{cluzet2020towards,cluzet2021croco} as described in \ citet{deschamps2021assimilation}. For each member, perturbations are auto-correlated in time following an auto-regressive process and are spatially homogeneous. The perturbation parameters were taken from \citet{charrois2016}. Precipitation parameters were adjusted (i.e. multiplicative noise with auto correlation time $\tau= 1500 h$, and dispersion $\sigma=0.5$) in order to obtain a spread-skill close to 1 for the open-loop run (see Sec. \ref{sec:res_ref}). We used these perturbed analyses as input for the snowpack simulations at the stations.

Thanks for pointing this out. We introduced some more elements. The following sentence was added:

For each member, perturbations are auto-correlated in time following an auto-regressive process and are spatially homogeneous.

- Line 151: Replace "radius" with "radii". OK

- Line 152: Should be "size" instead of "sizes". OK

- Line 156: Suggest replacing "remind" with "note". OK

- Line 157: Should be "rlocal". OK

- Line 166: Should be "requires use of independent data". OK

- Line 202: Add "of" at the end of this line. OK

- Line 203: Remove "If". OK

- Line 218: Replace "Despite" with "Although". OK

- Line 228: A final sentence is needed here to offer an explanation for interpreting the new formulation of the skill score.

The following sentence was added:

These properties are important to visually compare and average improvements (positive CRPSS) and degradations (negative CRPSS) of the CRPS.

- Line 236-237: This argument is not reliable, as the 2012 year looked to be snowier but the bias was lower (0.11 vs. 0.16). Please remove example or provide better justification/explanation.

This proposition was removed as it is not essential and would require unnecessary additional explanations. Thanks for pointing this out.

- Lines 239-240 and Table 1: Suggest adding a column with SS so the reader does not have to do the simple math.

The corresponding column was added to Table 1.

- Line 242: Add "negative" before "biases". OK

- Line 250: Replace "radius" with "radii". OK

- Line 292: Remove "Then". OK

- Line 295: Please revise the phrasing "pointed as", this is unclear to me. OK

'pointed as' was replaced by 'identified as'.

- Line 309: Please clarify which simulation is being referred to here when discussing the "simulation bias".

We referred to all the simulations. Rephrased into:

The most interesting feature here, is that the biases of all the simulations are increasing

- Line 316: Replace "inferior to" with "less than". OK

- Line 335: The phrasing is awkward here and I suggest rewording "enables to satisfactorily account for".

Reworded into:

The open-loop run is reliably accounting for its modelling uncertainties and errors, since its SS is slightly below unity over the ten years.

- Line 402: Replace "Despite" with "Although". OK

- Line 423: Delete "Then,". OK

- Line 431: Delete "of course". OK

**TABLE AND FIGURE COMMENTS**

- Figure 2 could be simplified into two panels: 2009-2015 and 2016-2018, since 2016 seems to be the most major change in the station network in the study period. Otherwise, the histograms appear to be similar. OK

 Thanks for this suggestion. Figs 2 and 3 were merged into one single figure, and winters 2011-2012 and 2017-2018 were selected for display:

[Figure]

**REFERENCES**

Clark, M. P., Hendrikx, J., Slater, A. G., Kavetski, D., Anderson, B., Cullen, N. J., et al. (2011). Representing spatial variability of snow water equivalent in hydrologic and land-surface models: A review. *Water Resources Research*, *47*(7), W07539. https://doi.org/10.1029/2011WR010745

Lundquist, J., Hughes, M., Gutmann, E., & Kapnick, S. (2019). Our Skill in Modeling Mountain Rain and Snow is Bypassing the Skill of Our Observational Networks. *Bulletin of the American Meteorological Society*, *100*(12), 2473–2490. https://doi.org/10.1175/BAMS-D-19-0001.1

Vernay, M., Lafaysse, M., Monteiro, D., Hagenmuller, P., Nheili, R., Samacoïts, R., Verfaillie, D., and Morin, S.: The S2M meteorological and snow cover reanalysis over the French mountainous areas, description and evaluation (1958–2020), Earth Syst. Sci. Data Discuss. [preprint], https://doi.org/10.5194/essd-2021-249, in review, 2021.

---

## Author Comment (AC2)

**RC2**: 'Comment on tc-2021-225', Anonymous Referee #2, 27 Sep 2021

The authors would like to thank the reviewer for his accute review asking for more clarity in the description of the procedure. Sec. 2 was considerable adjusted to fit with these requirements, and we believe that new Fig. 3 will really help understand our assimilation methods. Note that some slight changes were also made in the manuscript in order to improve its clarity, and are visible in the track changes. In the following, the reviewer initial comments are written in black, our answer in blue and the corrections in the paper are highlighted in red.

**Major comments**

In the paper titled "Propagating information from snow observations with CrocO ensemble data assimilation system: a 10-years case study over a snow depth observation network" Bertrand Cluzet et al. assimilated snow depth observations from an in-situ network of 295 stations covering the French Alps, Pyrenees, and Andorra over the period 2009–2019. They attempted to demonstrate how in-situ observations of snow depth can help contain intermediate and large-scale modeling errors by means of data assimilation. However, while the results of snow depth data assimilation are closer to observations than the non-operational open-loop counterpart (open-loop), which does not use data assimilation, they are not as good as the operational deterministic snow cover modeling system (oper). It is natural that the results of snow depth data assimilation are better than those of open-loop, but it is necessary to show that snow data assimilation outperforms oper. Moreover, I believe that snow depth data assimilation not only improves the reproducibility of the same snow depth but also affects other parameters of the snow model such as snow water equivalent, snow density, snow surface temperature, and water and energy balances above the snow; hence, I would like to see the discussion on the impact and verification of these factors. The procedure of the data assimilation method is difficult to understand, especially the need to clarify the control variables and the analysis variables. Overall, the paper is not accepted in the current form and needs to be revised in accordance with the following comments. My main concerns are as follows:

(1) Although the method of data assimilation has already been published in papers such as Cluzet et al. (2020, 2021), I feel that it is necessary to describe the flowchart of data assimilation and to clarify what the control variables and analysis values are, so that the data assimilation method can be understood to some extent from this study independently. I feel that since there is a limit to the extent of explanation that can be expressed only in text, I would really like to see a flowchart of data assimilation. I think that data assimilation can be corrected for other elements of the model using cross-covariance and error covariance with snow depth. It should be clarified whether snow depth is the only control variable. Furthermore, the authors conducted two types of experiments for data assimilation using particle filters, but I did not understand the difference between rlocal and klocal. Please explain this in detail

The authors want to thank the reviewer for this detailed question. We believe that the resulting adjustments will really improve the quality and clarity of the manuscript. We identify several elements in this question, and answer it point by point.

(a) need for a flowchart describing the control/analysis variables:

A new Section (2.2.4) was added, with a new figure (Fig. 3 of the revised manuscript), showing a simplified example of localized data assimilation with two points. Beyond the reference to Cluzet et al., (2021), we believe that this example can illustrate the fact that we use a 160 member ensemble, the sequential weekly assimilation procedure, and show how non-local observations can constrain the ensemble locally.

This section presents an illustrative example for the propagation of information with the localised PF. On December 3\textsuperscript{rd}, 2009, we perform an analysis at an unobserved point $p_{loc}$ (2135 \unit{m.a.s.l}) using an observation from a nearby point $p_{obs}$ (2293 \unit{m.a.s.l}, 7 \unit{km} away). The top panel of Fig. \ref{fig:example} shows the HS simulated by the 160 ensemble members at the two locations until the considered assimilation date. The observed HS at $p_{obs}$ is 0.87 \unit{m}, above the ensemble median at this location (about 0.5 \unit{m}). The PF will likely select the particles that have above average HS at $p_{obs}$. The bottom panel of Fig. \ref{fig:example} shows the particles' HS values at $p_{obs}$ as a function of their value at $p_{loc}$. A correlation can be noted: the particles predicting the highest HS at $p_{loc}$ usually also predict higher than average HS at $p_{obs}$. It means that the ensemble that we constructed (see Sec. \ref{sec:assim_setup}) considers that the modelling errors are linked: if there is an underestimated snowfall in early December at $p_{obs}$, it's likely that this is also the case at $p_{loc}$.\\

The localised PF performs an analysis for $p_{loc}$ by comparing the values modelled at $p_{obs}$ with the available observation, thereby selecting the 'best' particles at $p_{obs}$, (bottom panel, in green). The marginal distribution of the ensemble at $p_{obs}$ (right of the bottom panel, in green) is significantly sharpened compared to the background, and is much closer to the observation. At $p_{loc}$, the distribution of the HS values of these particles is also sharper, and exhibits higher HS than before the analysis.\\

This example shows how the localised PF has used the non-local observation at $p_{obs}$ to infer information about the local unobserved point $p_{loc}$. This example can be generalized to the situation where multiple observations are assimilated simultaneously as done in this study. It also highlights the implicit importance of ensemble correlations with distant locations: in the absence of correlation, no information can be transferred. In such a situation, the klocal algorithm would discard the observations from the least areas, while the rlocal would keep them. Finally, note that if the ensemble correlation is dramatically wrong, (i.e. positive correlation instead of negative correlation), the analysis will degrade the ensemble performance.\\

(b) The method should be understood to some extent from this study independently.

Sec. 2.2.3 has been completely re handled, in order to give more details and explanations on the data assimilation flowchart.

The Particle Filter used in this work is based on the version described in \citet{cluzet2021croco}. Only a brief description of the procedure is given here. The ensemble is updated sequentially with the PF on each assimilation date and propagated forward until the following assimilation date. The PF is localised: each point receives a different analysis. Based on the comparison of neighbouring simulations of HS with their corresponding HS observations, the PF selects a sample of the best ensemble members. The idea is that if a particle is performing well against nearby observations, it should also be efficient locally \citep{farchi2018comparison}. Different localisation radius are tested in this study ranging from 17 \unit{km} to 300 \unit{km}. Note that when a particle is selected by the PF, the full local state vector is copied: the local physical consistency of the variables is preserved.\\

Particle Filter degeneracy (see Sec. \ref{sec:intro}) may arise even with a reduced local domain size, and approaches to increase the PF tolerance may be required to overcome it. The localisation is complemented here by two different strategies described in \citet{cluzet2021croco}, inflation and k-localisation, leading to the 'rlocal' and 'klocal' algorithms, respectively. If the initial analysis is degenerated (i.e. the effective sample size $N_{eff}$ is inferior to a target $N_{eff}^*$), the rlocal and klocal iteratively modify the assimilation settings to make it more tolerant, so that the PF analysis reaches a sample size of $N_{eff}^*$. The rlocal algorithm performs an inflation of observation errors inspired by \citet{larue2018assimilation}. The klocal algorithm discards observations coming from locations exhibiting the lower ensemble correlations with the considered location. It is important to note that inside a localisation radius, the rlocal method assimilates all available observation stations whereas the klocal method only selects a subset of observations from locations where the ensemble members are sufficiently correlated with the simulation members of the considered point.\\

(c) I think that data assimilation can be corrected for other elements of the model using cross-covariance and error covariance with snow depth.

With the answer to (b), it should be clear now that in the PF, the full local state vector is replicated, not only HS. Therefore no statistical assumption is made to update other state variables from HS.

(d) It should be clarified whether snow depth is the only control variable.

HS is unambiguously the only assimilated variable (in the submitted manuscript, l. 4-5, 10-12, 16-18, 53-55, 89-97). This ambiguity should be raised with the answers to points (a) and (b).

(c) explain in detail the difference between the rlocal and the klocal approaches. The difference between the two algorithms were described in l.158-164 of the manuscript (Sec. 2.2.3). This paragraph was rewritten in order to improve its clarity and details were added.

Particle Filter degeneracy (see Sec. \ref{sec:intro}) may arise even with a reduced local domain size, and approaches to increase the PF tolerance may be required to overcome it. The localisation is complemented here by two different strategies described in \citet{cluzet2021croco}, inflation and k-localisation, leading to the 'rlocal' and 'klocal' algorithms, respectively. If the initial analysis is degenerated (i.e. the effective sample size $N_{eff}$ is inferior to a target $N_{eff}^*$), the rlocal and klocal iteratively modify the assimilation settings to make it more tolerant, so that the PF analysis reaches a sample size of $N_{eff}^*$. The rlocal algorithm performs an inflation of observation errors inspired by \citet{larue2018assimilation}. The klocal algorithm discards observations coming from locations exhibiting the lower ensemble correlations with the considered location. It is important to note that inside a localisation radius,

the rlocal method assimilates all available observation stations whereas the klocal method only selects a subset of observations from locations where the ensemble members are sufficiently correlated with the simulation members of the considered point.\\

(2) I would like to see an explanation of the difference between Oper and open-loop as a reference.

The distinction between oper and openlop was briefly given at the beginning of Sec. 3 (l. 161-164 of the submitted manuscript). This paragraph was updated to improve its clarity:

This work aims at assessing the potential transfer of information between points in an HS observation network by means of localized data assimilation, and more specifically to address the questions presented in the end of Sec. \ref{sec:intro}. To demonstrate that, the data assimilation system must over-perform its ensemble counterpart with the assimilation switched off (open-loop) and the state-of-the-art operational deterministic snow cover modelling system from Météo-France (oper), which consists in a default Crocus version forced by the unperturbed SAFRAN meteorological forcing \citep{vernay2021meteorological}.

I would also like to see a more detailed explanation of the SAFRAN massif, with expanded abbreviation, and the convincing data provided by SAFRAN to the snow model.

Details on SAFRAN massifs were given in Sec. 2.2.2. We aknowledge that this lacked some clarity, and thank the reviewer for pointing this out. The first paragraph of this section was expanded, insisting in particular on how SAFRAN analysis is performed over each massif, and then interpolated at the stations of the study:

Meteorological forcings are taken from SAFRAN (Système D'Analyse Fournissant des Renseignements Adaptés à la Neige, \citet{durand1993}) reanalysis over the Alps and Pyrenees. SAFRAN is a surface meteorological analysis system adjusting backgrounds from NWP model ARPEGE \citep{courtier1991} with local meteorological observations (air temperature, pressure, precipitation, humidity) within so-called massifs of about 1000 \unit{km^2} (see Fig. \ref{fig:meth_obs_density_massif}) and further downscaled to the stations of our study. Over the considered period of time, 438 observation sites provided precipitation observations to SAFRAN between November and April. These stations are mostly located at lower elevations (below 1500 m) as presented in Fig.4 of \citet{vernay2021meteorological}. Among them, 164 of these sites correspond to locations with snow depth observations included in the present study. SAFRAN analysis is issued separately for each massif in a semi-distributed geometry, i.e within 300 \unit{m} elevation bands, aspect and slopes, the main topographic parameters controlling the snow cover evolution. This analysis is subsequently downscaled into the specific topographic conditions (i.e. elevation, slope, aspect and local topographic mask) of the simulated station \citep{vionnet2016}. This means that a same analysis is applied to all the points within a same massif, and interpolated consistently with their topographic parameters, while analyses for neighbouring stations located in distinct massifs will be different.\\

(3) Advantage of data assimilation is that by assimilating snow depth data, the information is correlated to other factors such as snow water content and precipitation, which will improve the accuracy of model estimation other than direct snow depth improvement. If you have observational data on snow density, snow surface temperature and snow water content, please add a discussion on whether the assimilation of snow depth affects other model parameters.

Such data is unfortunately unavailable at the studied stations. We agree that this would have been a valuable source of information for this study.

(4) I feel that there is a lack of literature in the introduction. As a pioneering work in satellite data assimilation, I feel it is necessary to mention the data assimilation study of MODIS snow cover and AMSR-E snow water content by Andreadis and Lettenmaier (2006).

We agree that this pioneering work should have been cited in the introduction. The sentence corresponding to l. 38-41 was adapted to include this reference:

Andreadis, K. M.; Lettenmaier, D. P. Assimilating remotely sensed snow observations into a macroscale hydrology model. Adv. Water Resour. 2006, 29, 872–886.

Data assimilation of snowpack observations may help address this issue in complement to these observations. Remotely-sensed retrieval of snow bulk properties (e.g. the height of snow (HS, \unit{m}) and the snow water equivalent (SWE, \unit{kg} \unit{m^{-2}})) is a promising wealth of snowpack observations for data assimilation \citep[e.g.][] {margulis2019utility} but it is inherently limited by spatio-temporal gaps \citep{delannoy2012}, or only available at coarse resolutions \citep{andreadis2006assimilating}.

Line 42–43: "Their potential to improve local simulations is unambiguous as demonstrated by many studies." After this sentence, please consider adding references such as Liston and Heimstra (2008) and Suzuki et al. (2015).

Liston, G. E.; Hiemstra, C. A. Simple data assimilation system for complex snow distributions (SnowAssim) J. Hydrometeorol. 2008, 9, 989–1004.

Suzuki, K.; Liston, G. E.; Matsuo, K. Estimation of continental-basin-scale sublimation in the Lena River Basin, Siberia. Adv. Meteorol. 2015, 2015.

The authors would like to thank the reviewer for these interesting citations. A reference to Liston et Heimstra (2008), was added in the following sentence, because it was more fair for this citation (which is more ambitions than only improving local simulations)

The potential to transfer information into neighbouring areas is therefore a key question when considering their potential added value for snow cover modelling over large domains \citep[e.g.][]{slater2006snow,liston2008simple,gichamo2019ensemble}

Line 61–62: "These variants are used in a localised framework, in which only observations coming from a certain radius around the considered location are assimilated." After this, please consider adding references such as Zupanski (2021).

Zupanski, M. The Maximum Likelihood Ensemble Filter with State Space Localization. *Mon. Weather Rev.* 2021.

With the corresponding sentence, we wanted to focus on particle filter Particle Filter localization methods that could directly relate to our approach. Even though this brand new method may have similarities with the PF method, we find it a bit off topic (ery theoretical) here.

(5) Line 389: There are two reasons why the operational run could not be beaten by the assimilation in terms of RMSE.

Reworded into:

There are two reasons why the assimilation could not outperform the operational run in terms of RMSE.

Line 454–455: In other words: the assimilation can not beat the open-loop in the most densely observed areas (e.g. in the Northern Alps, 455 where the observation density is similar to the studies of Magnusson et al. (2014) and Winstral et al. (2019)) because the open-loop performance is already high there.

Here, 'beat' was replaced by 'outperform'

In the above two sentences, the verb "beat" is used, but I feel it is not appropriate. I would like to see this correction with more appropriate words.

Line 267–269: The text needs to be revised and made more readable.

Reworded into:

In the following, we will investigate the different factors influencing the skill variability of the assimilation runs. As described in the previous Sec. \ref{sec:res_assim}, there are only small skill differences between the localised radii of 17-50 \unit{km}, and between the rlocal and klocal algorithm. For the sake of illustration, we decided to focus on the assimilation configuration yielding the lowest median RMSE. This configuration, the klocal with a 35 \unit{km} localisation radii, is further referred to as 'klocal' configuration.

(6) The results of data assimilation in Fig. 6 should be shown in the form of Table 1 with numerical values. Moreover, there is no data for oper in RMSE and SS in Fig. 6; please resolve this.

The authors would like to thank the reviewer for suggesting an improvement to this figure. There is actually a flaw, with a missing RMSE plot for th oper. This was corrected in the revised version of the manuscript. However, the ss cannot be computed for the oper (since it is a deterministic run, there is no spread), this is why ss is missing. And we are not sure whether putting all the data in a table would be readable, as there would be five times more data than in Table 1.

Added RMSE values for the open on Fig. 6.

References:

Cluzet, B., Lafaysse, M., Cosme, E., Albergel, C., Meunier, L.-F., and Dumont, M.: CrocO_v1. 0: a particle filter to assimilate snowpack observations in a spatialised framework, Geoscientific Model Development, 14, 1595–1614, https://doi.org/10.5194/gmd-14-1595-2021, 2021.

---

## Author Comment (AC3)

**RC3**: 'Comment on tc-2021-225', Anonymous Referee #3, 30 Sep 2021

The authors would like to thank the reviewer for his acute review asking for more clarity in the description of the procedure, globally in the same line as RC2. Sec. 2 was considerable adjusted to fit with these requirements, and we believe that new Fig. 3 will really help understand our assimilation methods. As special care was taken to the quality of the figures. Note that some slight changes were also made in the manuscript in order to improve its clarity, and are visible in the track changes. In the following, the reviewer initial comments are written in black, our answer in blue and the corrections in the paper are highlighted in red.

The paper demonstrates that assimilation of in-situ sown depth observations does not provide significant RMSE improvements over the open-loop and operational simulations. It reduces bias in snow depth estimates and outperforms the open-loop simulations in specific elevation bands in locations with a lower observation density.

The methodology and the presentation of results are clear, and I agree that this approach seems to be relevant for the estimations of snow depth and SWE. For all these reasons, I think the paper should be published with a minor revision.

The paper is interesting and provides insight into analyzing ensemble data assimilation approaches. However, there is too much emphasis on the Continuous Ranked Probability Score (CRPS).  It would be more insightful to show snow depth maps or scatter plots of model simulations (open loop, oper, DA vs. observations).

Thanks for this thorough remark on our methodology and presentation of results. Indeed, focusing on the CRPS is a deliberate choice, as it is much more suited to the evaluation of ensembles than RMSE or bias, since it accounts for the actual distribution of the ensembles, and offers a way to aggregate scores both spatially and temporally, giving robustness to the evaluations. RMSE and bias are given as a means to compare the ensemble median to the operational (deterministic) run globally (Tab. 1 and Fig. 6). Snow depth maps cannot be plotted, as simulations are only ran at the stations, and not in a grided setting. We also argue that scatter plots would lack the ability to synthesize information, while requiring the need to reduce the ensembles to their median. However, we acknowledge that illustrations on the behaviour of the ensemble (in terms of spread, and temporal variations) would be appreciated, and therefore we think that Fig. 3 of the revised manuscript provides some insight on the ensemble behavior.

[Figure]

L137-140: Please provide perturbation statistics (additive/multiplicative, mean, standard deviations/correlation coefficients, spatial, and temporal correlations) for each forcing).

We agree that the lack of details was detrimental to the clarity of the study. However, we believe that explaining the full perturbation procedure, already described in the provided reference, would help focusing on the meaningful details. We rather added some outstanding details. In particular we point out that perturbations are spatially homogeneous (see below)

An ensemble of forcings was generated by applying stochastic perturbations in the same spirit as \citet{charrois2016} but with slight corrections in the implementation of the perturbations compared with \citet{cluzet2020towards,cluzet2021croco} as described in \citet{deschamps2021assimilation}. For each member, perturbations are auto-correlated in time following an auto-regressive process and are spatially homogeneous. The perturbation parameters were taken from \citet{charrois2016}. Precipitation parameters were adjusted (i.e. multiplicative noise with auto correlation time $\tau= 1500 h$, and dispersion $\sigma=0.5$) in order to obtain a spread-skill close to 1 for the open-loop run (see Sec. \ref{sec:res_ref}). We used these perturbed analyses as input for the snowpack simulations at the stations.

L165: a summary about the updating step and how the PF updates the snow profile would improve the clarity of the setup section.

The authors would like to thank the reviewer for pointing our the lack of clarity and details of the corresponding Sec. 2.2.3. This section was thoroughly rewritten. Furthermore, a new Sec. 2.2.4 was added to illustrate the behavior of the localised POF in a simple example. We belive that this will really improve the understandably of the manuscript.

The Particle Filter used in this work is based on the version described in \citet{cluzet2021croco}. Only a brief description of the procedure is given here. The ensemble is updated sequentially with the PF on each assimilation date and propagated forward until the following assimilation date. The PF is localised: each point receives a different analysis. Based on the comparison of neighbouring simulations of HS with their corresponding HS observations, the PF selects a sample of the best ensemble members. The idea is that if a particle is performing well against nearby observations, it should also be efficient locally \citep{farchi2018comparison}. Different localisation radius are tested in this study ranging from 17 \unit{km} to 300 \unit{km}. Note that when a particle is selected by the PF, the full local state vector is copied: the local physical consistency of the variables is preserved.\\

Particle Filter degeneracy (see Sec. \ref{sec:intro}) may arise even with a reduced local domain size, and approaches to increase the PF tolerance may be required to overcome it. The localisation is complemented here by two different strategies described in \citet{cluzet2021croco}, inflation and k-localisation, leading to the 'rlocal' and 'klocal' algorithms, respectively. If the initial analysis is degenerated (i.e. the effective sample size $N_{eff}$ is inferior to a target $N_{eff}^*$), the rlocal and klocal iteratively modify the assimilation settings to make it more tolerant, so that the PF analysis reaches a sample size of $N_{eff}^*$. The rlocal algorithm performs an inflation of observation errors inspired by \citet{larue2018assimilation}. The klocal algorithm discards observations coming from locations exhibiting the lower ensemble correlations with the considered location. It is important to note that inside a localisation radius, the rlocal method assimilates all available observation stations whereas the klocal method only selects a subset of observations from locations where the ensemble members are sufficiently correlated with the simulation members of the considered point.\\

L229:  Please provide some information about the resolution of simulations.

How are the bias and RMSE computed (i.e., point vs. gridded simulations)? What is the uncertainty of this comparison?

The authors can elaborate more on the representative error. Gridded simulations are compared with ground-based point measurements. There is a scale gap in this comparison. It will benefit the paper if the authors discuss this source of uncertainty.

The authors acknowledge that more details were necessary in the description of the simulation setup. The simulations are not gridded, but performed at point locations colocated with the measurements: there is no scale mismatch between simulations and observations, and therefore no uncertainty in the comparison. We agree that there remains representativeness errors in the observations, because of snowpack varibility at the plot scale scale.

We also clarified the model configuration by stating more clearly that SAFRAN analyses are downscaled at the stations in Sec. 2.2.2:

Meteorological forcings are taken from SAFRAN reanalysis over the Alps and Pyrenees. SAFRAN \citep{durand1993} is a surface meteorological analysis system adjusting backgrounds from NWP model ARPEGE \citep{courtier1991} with local meteorological observations (air temperature, pressure, precipitation, humidity) within so-called massifs of about 1000 \unit{km^2} (see Fig. \ref{fig:meth_obs_density_massif}) and further downscaled to the stations of our study. [...] SAFRAN analysis is issued separately for each massif in a semi-distributed geometry, i.e within 300 \unit{m} elevation bands, aspect and slopes, the main topographic parameters controlling the snow cover evolution. This analysis is subsequently downscaled into the specific topographic conditions (i.e. elevation, slope, aspect and local topographic mask) of the simulated station \citep{vionnet2016}.

We also adapted the discussion  in Sec 5.4 was rewritten to reflect the consideration on potential scale mismatch and observation representativeness:

Nevertheless, obtaining a perfect spread-skill may be a challenging goal for our assimilation system. Indeed, under dispersion is a common issue in the NWP \citep[e.g.][]{bellier2017sample} and snow cover modelling communities \citep{lafaysse2017multiphysical,nousu2019statistical}, and can be explained here by several factors. On the one hand, despite meteorological forcings are downscaled to the stations (see Sec.  \ref{sec:forc}, so that there is no scale mismatch), the ensemble modelling chain does not account for two important processes affecting the observations. The variability of the meteorological conditions inside SAFRAN massifs is limited to topographic parameters (including local masks) so that two distant stations with the same topography will receive the exact same forcing, and the snow redistribution by wind is not represented \citep{vionnet2018,mott2018seasonal}. On the other hand, the representativeness of observations is limited by plot-scale variability.
\\
Data assimilation is known to partly compensate for such mismatches via error compensation \citep{klinker1992diagnosis,rodwell2007using,wong2020model}. For example, an ablation event in one observation can be compensated in the Particle Filter by selecting some members with a lower precipitation factor or a compaction scheme with a higher settling \citep{deschamps2021assimilation}. This compensation immediately results in lower errors, but implicitly, the model does a wrong assumption, which results in being over confident, thus with a lower spread. The only way to mitigate for this over confidence is to account for any relevant physical phenomenon, which is a desirable goal, but a real challenge when it comes to snowdrift by wind, local meteorology and plot-scale variability. This goal is to date out of reach at the temporal and spatial scale of this study.\\

L300: What is the reason for presenting figure 10b?

This figure is  presented and commented in l. 298-301 of the submitted manuscript. The intention is to show that there is a relation between the openloop bias and the density of observations.

If possible, please improve the quality of the figures.

Thanks for this remark. More than half of the figures were carefully rehandled (Figs. 1, 2, 4, 5, 7, 8 and 11).